# Improving drug discovery using image-based multiparametric analysis of the epigenetic landscape

Chen Farhy[1], Santosh Hariharan[2,3], Jarkko Ylanko[2,3], Luis Orozco[1], Fu-Yue Zeng[1], Ian Pass[1], Fernando Ugarte[4,5], E Camilla Forsberg[4,5], Chun-Teng Huang[1], David W Andrews[2,3,6], Alexey V Terskikh[1]*

[1]Sanford Burnham Prebys Medical Discovery Institute, La Jolla, United States; [2]Biological Sciences Platform, Sunnybrook Research Institute, University of Toronto, Ontario, Canada; [3]Department of Medical Biophysics, University of Toronto, Ontario, Canada; [4]Department of Biomolecular Engineering, University of California, Santa Cruz, Santa Cruz, United States; [5]Institute for the Biology of Stem Cells, University of California, Santa Cruz, Santa Cruz, United States; [6]Department of Biochemistry, University of Toronto, Ontario, Canada

*For correspondence:
terskikh@sbpdiscovery.org

Competing interests: The authors declare that no competing interests exist.

**Abstract** High-content phenotypic screening has become the approach of choice for drug discovery due to its ability to extract drug-specific multi-layered data. In the field of epigenetics, such screening methods have suffered from a lack of tools sensitive to selective epigenetic perturbations. Here we describe a novel approach, Microscopic Imaging of Epigenetic Landscapes (MIEL), which captures the nuclear staining patterns of epigenetic marks and employs machine learning to accurately distinguish between such patterns. We validated the MIEL platform across multiple cells lines and using dose-response curves, to insure the fidelity and robustness of this approach for high content high throughput drug discovery. Focusing on noncytotoxic glioblastoma treatments, we demonstrated that MIEL can identify and classify epigenetically active drugs. Furthermore, we show MIEL was able to accurately rank candidate drugs by their ability to produce desired epigenetic alterations consistent with increased sensitivity to chemotherapeutic agents or with induction of glioblastoma differentiation.

## Introduction

The epigenetic landscape of a cell is largely determined by the organization of its chromatin and the pattern of DNA and histone modifications. These confer differential accessibility to areas of the genome and through direct and in-direct regulation of all DNA-related processes, form the basis of the cellular phenotype (*Jenuwein and Allis, 2001*; *Lawrence et al., 2016*; *Berger, 2007*; *Goldberg et al., 2007*). By collecting global information about the epigenetic landscape, for example using ATAC- or histone ChIP-seq, we can derive multilayered information regarding cellular states (*Miyamoto et al., 2018*; *Mikkelsen et al., 2007*). These include stable cell phenotypes such as quiescence, senescence, or cell fate, as well as transient changes such as those induced by cytokines and chemical compounds. However, current methods for collecting such information are not adapted for high-content drug screening. Over the past decade the decreasing cost and remarkable scalability of high content screening have made it a particularly attractive alternative for drug discovery. More recently, novel image processing tools coupled with multiparametric analysis and machine learning have significantly impacted our ability to investigate and understand the output of phenotypic screens (*Kang et al., 2016*; *Scheeder et al., 2018*). Despite these advantages, such assays have not been adapted to extract and utilize information form the cellular epigenetic landscape.

**eLife digest** Each cell contains a complete copy of a person's genes coded in their DNA. However, for a cell to perform its specific role, it only needs a small fraction of this genetic information. The mechanisms that control which genes a cell is using fall under the umbrella of 'epigenetics' (meaning beyond genetics). These mechanisms involve changes in the way that DNA is organized inside the cell nucleus and changes in how accessible different parts of the genome are to various cellular components.

DNA is long and fragile so, to maintain its integrity, it is wrapped around protein complexes called histones. Adding chemical modifications to histones is one of the main epigenetic mechanisms that cells use to regulate which genes are turned on and off. Several methods allow researchers to read patterns of histone modification and use this information to derive what state a cell is in and how it might behave. Improving these methods is of particular interest in drug development, where this information could reveal the effects, and side-effects, of new treatments. Unfortunately, existing techniques are costly in both time and money, and they are not well suited to analyzing epigenetic changes caused by the large numbers of compounds tested during drug development.

To overcome this barrier, Farhy et al. have developed a new system called 'Microscopic Imaging of the Epigenetic Landscape' (MIEL for short). The system allows them to quickly analyze the epigenetic changes caused by each of a large number of different chemical compounds when they are used on cells. MIEL tags different histone modifications by staining each with a different color, and then uses automated microscopy to produce images. It then extracts information from these images using advanced image analysis tools. The changes induced by different drugs can then be compared and categorized using machine learning algorithms.

To test the MIEL system, Farhy et al. grew brain cancer cells (derived from human tumors) in the lab, and treated them with compounds that target proteins involved in histone modifications. Using their newly created pipeline, Farhy et al. were able to identify the unique epigenetic changes caused by these compounds, and train the system to correctly predict which type of drug the cells had been treated with. In a different set of experiments Farhy et al. demonstrate the utility of their new pipeline in identifying drugs that induce a set of epigenetic changes associated with a reduced ability to regrow tumors.

This new system could help screen thousands of compounds for their epigenetic effects, which could aid the design of new treatments for many diseases.

While malignant glioblastoma is the most common and lethal brain tumor, current therapeutic options offer little prognostic improvement, so the median survival time has remained virtually unchanged for decades (*Jhanwar-Uniyal et al., 2015*; *Parvez, 2008*; *Burger and Green, 1987*). Tumor-propagating cells (TPCs) are a subpopulation of glioblastoma cells with increased tumorigenic capability (*Patel et al., 2014*) operationally defined as early-passaged (<15) glioblastoma cells propagated in serum-free medium (*Nakano et al., 2008*). Compared to the bulk of the tumor, TPCs are more resistant to drugs, such as temozolomide (TMZ) and radiation therapy (*Bao et al., 2006*; *Safa et al., 2015*). This resistance may explain the failure of traditional therapeutic strategies based on cytotoxic drugs targeting glioblastoma. Multiple approaches aimed at reducing or circumventing the resilience of TPCs have been proposed. These include targeting epigenetic enzymes (i.e., enzymes that write, remove, or read DNA and histone modifications) to increase sensitivity to cytotoxic treatments (*Jones et al., 2016*; *Strauss and Figg, 2016*; *Lee et al., 2017*; *Romani et al., 2018*); and differentiating TPCs to reduce their tumorigenic potential (*von Wangenheim and Peterson, 1998*; *Von Wangenheim and Peterson, 2001*; *von Wangenheim and Peterson, 2008*; *Lee et al., 2015*; *Song et al., 2016*; *Garros-Regulez et al., 2016*).

Here, we introduce Microscopic Imaging of the Epigenetic Landscape (MIEL), a novel high-content screening platform that profiles chromatin organization using the endogenous patterns of histone modifications present in all eukaryotic cells. We validate the platform across multiple cell lines and drug concentrations demonstrating its ability to classify epigenetically active compounds by molecular function, and its utility in identifying off-target drug effects. We show MIEL can accurately

rank candidate drugs by their ability to produce a set of desired epigenetic alterations such as glioblastoma differentiation.

## Results

### The MIEL platform

We have developed a novel phenotypic screening platform, Microscopic Imaging of Epigenetic Landscape (MIEL), which interrogates the epigenetic landscape at both population and single cell levels using image derived features and machine learning. MIEL takes advantage of epigenetic marks such as histone methylation and acetylation, which are always present in eukaryotic nuclei and can be revealed by immunostaining. MIEL analyzes the immunolabeling patterns of epigenetic marks using conventional image analysis methods for nuclei segmentation, feature extraction, and previously described machine-learning algorithms (*Collins et al., 2015*) (*Figure 1a* and Materials and methods). Primarily, we utilized four histone modifications: H3K27me3 and H3K9me3, which are associated with condensed (closed) facultative and constitutive heterochromatin, respectively; H3K27ac, associated with transcriptionally active (open) areas of chromatin, especially at promoter and enhancer regions; and H3K4me1, associated with enhancers and other chromatin regions (*Figure 1a*; *Creyghton et al., 2010*; *Shlyueva et al., 2014*). To focus on the intrinsic pattern of epigenetic marks, we use only texture-associated features (e.g., Haralick's texture features [*Haralick et al., 1973*], threshold adjacency statistics, and radial features [*Hamilton et al., 2007*]) for multivariate analysis. Previous studies have successfully employed similar features for cell painting techniques combined with multiparametric analyses to accurately classify subcellular localization of proteins (*Hamilton et al., 2007*), cellular subpopulations (*Loo et al., 2009*), and drug mechanisms of action (*Collins et al., 2015*; *Caie et al., 2010*; *Gustafsdottir et al., 2013*; *Loo et al., 2007*).

We employed three main methods of data visualization and analysis: (1) To visualize similarity between multiple cell populations across all acquired features we conducted multidimensional scaling (MDS) using the Euclidean distance between the multivariate centroids of all populations being compared and displayed the results as a 2D scatter plot (termed distance map; Materials and methods and *Figure 1a*). (2) To classify multiple cell populations, we employed quadratic discriminant analysis of multivariate centroids(DA; Materials and methods and *Figure 1a*). (3) Single cells within each cell populations were classified using a Support Vector Machine (SVM; Materials and methods and *Figure 1a*).

The most commonly used cellular assays for epigenetic drug discovery are lysis and ELISA based assays, such as AlphaLISA (PerkinElmer). Imaging-based alternatives rely on staining for relevant histone modification and monitoring changes in average fluorescent intensity (*Sayegh et al., 2013*; *Luense et al., 2015*). Using MIEL, we screened a library of 222 epigenetically active compounds, many with known targets among epigenetic writers, erasers, or readers (SBP epigenetic library, *Figure 1—figure supplement 1a,b*). Our analysis focused on MIEL's ability to (1) detect active compounds; (2) group drugs by function and identify off-target effects; (3) be robust across cell lines and drug concentrations; (4) rank active drugs, and derive information regarding drug mechanism of action.

### MIEL improves detection of epigenetically active drugs

To test the ability of the MIEL approach to detect active compounds and compare it to intensity-based methods, primary-derived TPCs (GBM2 cell line) were treated with epigenetically active drugs for 24 hr (10 μM, triplicates). Treated cells were immuno-labeled for multiple histone modifications expected to exhibit alterations following drug treatment (H3K9me3, H3K27me3, H3K27ac and H3K4me1). Image analysis including nuclei segmentation and features extraction was conducted, as previously described (*Collins et al., 2015*) on Acapella 2.6 (PerkinElmer). Phenotypic profiles were generated for each compound or control (DMSO) treated wells. These are vectors composed of 1048 (262 features per modification X four modifications) texture features derived from the staining of each histone modification and representing the average value for each feature across all stained cells in each cell population (drug or DMSO). When treatment reduced cell count to under 50 imaged nuclei per well, the compound was deemed toxic and excluded from analysis. Following feature normalization by z-score, we calculated the Euclidean distance between vectors of the

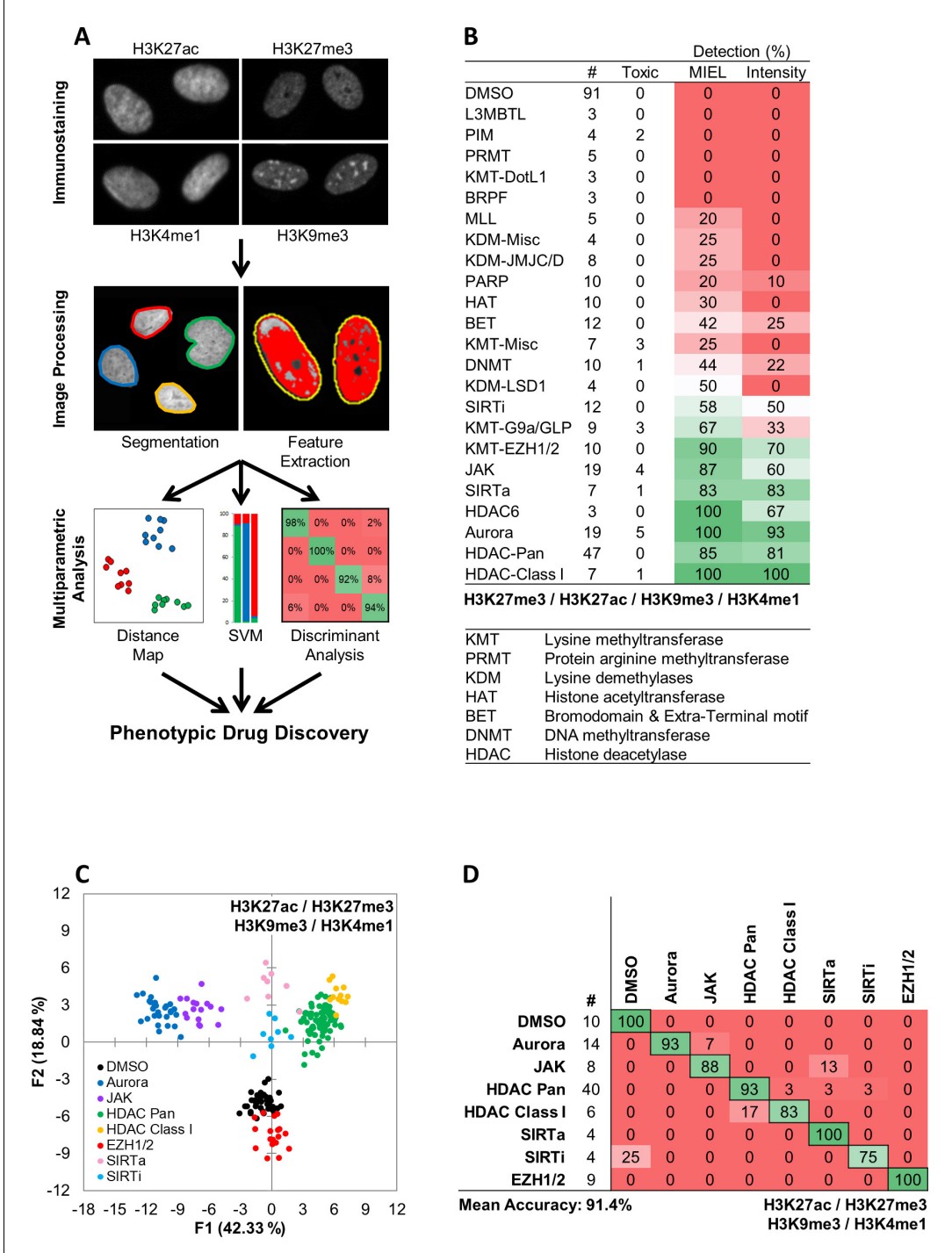

**Figure 1.** MIEL compares the epigenetic landscape of multiple cell populations and can be used to detect active epigenetic drugs across cell lines and drug concentrations. (A) Flowchart of MIEL pipeline. Fixed cells were immunostained for the desired epigenetic modifications and imaged. Nuclei were segmented based on DNA staining (Hoechst 33342 or DAPI) and texture features were calculated from the pattern of immunofluorescence. The relative similarity of multiple cell populations was assessed by calculating the multi-parametric Euclidean distance between populations centers, and represented in 2D following MDS (distance map). Discriminant analysis is used to functionally classify whole cell populations based on their multi-parametric centers. SVM classification is used to separate single cells in each population and estimate populations overlap. (B) Table showing the fraction of epigenetic drugs in each functional category identified as active by either MIEL analysis employing texture features derived from images of GBM2 cells stained for H3K9me3, H3K4me1, H3K27ac, H3K27me3, or by intensity-based analysis using the same modifications (see Materials and methods). (C,D) Quadratic discriminant analysis using texture features derived from images of GBM2 cells treated with either DMSO or 85 active compounds (two technical replicates per compound; 38 DMSO replicates) stained for H3K9me3, H3K27me3, H3K4me1, H3K27ac. (C) Scatter

*Figure 1 continued on next page*

*Figure 1 continued*

plots depicting the first two discriminant factors derived from features of all four histone modification images for each cell population. (D) Confusion matrix showing classification results of discriminant analysis. Left column details number of compounds or DMSO replicates for each category in the test set (one replicate per compound). Numbers represent the percent of compounds classified correctly (diagonal) and incorrectly (off the diagonal). The online version of this article includes the following figure supplement(s) for figure 1:

**Figure supplement 1.** Overview of the epigenetic drug library used in this study.
**Figure supplement 2.** MIEL improves detection rates and effect size compared to intensity based methods.
**Figure supplement 3.** Intensity based functional classification shows reduced classification accuracy compared with MIEL.
**Figure supplement 4.** MIEL can distinguish between HDAC inhibitors with different specificity.
**Figure supplement 5.** Low concentration valproic acid treatment induces epigenetic and transcriptomic changes distinct from that of known HDAC inhibitors.

compounds and DMSO- treated cells. These distances were then normalized (z-score) to the average distance between DMSO replicates and the standard deviation of these distances. Compounds with a distance z-score of greater than three were defined as active (see Materials and methods section). This analysis identified 122 compounds that induced significant epigenetic changes. Active compounds were not uniformly distributed across all functional drug categories. Rather, we identified 10 categories in which 50% of the drugs were categorized as active and nontoxic and 13 categories in which 25% or fewer of the drugs induced detectable epigenetic alterations following a 24 hr treatment (*Figure 1b*).

To compare MIEL with current thresholding methods, we repeated the calculation using mean fluorescence intensity for all histone modifications by normalizing (z-score) each drug against DMSO; active compounds were defined as compounds for which z-scored intensity for at least one of the histone modifications was greater than three or less than $-3$. As a result, we identified 94 active compounds, which were distributed across functional categories similarly to MIEL-identified compounds (*Figure 1b*). For each functional category, the number of compounds identified as active using thresholding was smaller than the number identified using MIEL (*Figure 1b*) demonstrating MIEL's increased detection sensitivity over standard thresholding.

To determine the contribution of individual histone modifications, we repeated both MIEL and thresholding analyses individually for each of the four modifications. Using MIEL-based analysis, a single modification yielded similar detection rates to the combination of modifications across most functional categories (*Figure 1—figure supplement 2a*). Using intensity-based analysis, individual modifications yielded lower detection rates compared to the combination of modifications and displayed equal or reduced detection rates when compared to MIEL in all categories and modifications (*Figure 1—figure supplement 2a*). Of note, 3 of the four modifications used for MIEL analysis showed similar detection rates across most of the functional categories. However, detection rates using H3K27me3 were consistently reduced across most active categories (*Figure 1—figure supplement 2a*) except for EZH1/2 inhibitors, possibly due to the role these enzymes play in regulating this posttranslational modification. To further compare MIEL and thresholding, we estimated the magnitude of epigenetic alterations induced by active compounds. We calculated the fold increase in distance from DMSO (normalized to average distance between DMSO replicates) as well as the fold change in fluorescence intensity for active compounds in each category. In all categories, MIEL showed an increased effect (*Figure 1—figure supplement 2b*).

These results demonstrate that, across all tested epigenetic marks, detecting epigenetically active compounds was markedly improved using the MIEL method compared to current image-based thresholding methods.

## MIEL correctly classifies epigenetic compounds by function and detects off-target effects

One key advantage of phenotypic profiling methods like MIEL is the ability to classify compounds by function and identify nonspecific effects through comparison with profiles of well-defined controls. To assess whether MIEL could correctly group compounds by function, we applied discriminant analysis (DA) to all active, nontoxic compounds from categories that had at least three such compounds (85 compounds; seven categories and DMSO). Two replicates from each drug and 38 DMSO replicates were used as a training set for a quadratic DA, using all texture features derived from images

of the four histone modifications. The third replicate for each compound, as well as 10 DMSO replicates, was used as a test set to validate the model. Results showed that MIEL separated multiple categories of epigenetically active drugs with an average accuracy of 91.4% (*Figure 1c,d*). Although many of the epigenetically active compounds induced alterations in average fluorescence (*Figure 1—figure supplement 2b*), a DA utilizing intensity measurements from all four channels was ineffective at separating the various categories and yielded only 51.6% average accuracy (*Figure 1—figure supplement 3a*). To test whether individual histone modification textures contain sufficient information to distinguish between the various drug classes, we performed DA using features derived from each modification. Although this degraded MIEL's ability to separate compound subclasses, which affected similar changes in histone modification such as Class I and Pan HDAC inhibitors, MIEL was still able to separate major categories, such as histone phosphorylation and deacetylation (*Figure 1—figure supplement 3b*).

DNA labeling dyes such as DAPI and Hoechst can partially recapitulate the staining pattern of H3K9me3, which labels constitutive heterochromatin. To test the ability of DNA labeling dyes to capture information regarding chromatin organization and their usefulness for function based classification, we used texture features derived from the DAPI channel to repeat the functional classification. This yielded an overall classification accuracy of 65.6% (*Figure 1—figure supplement 3b*) compared with 86.4% provided by H3K9me3 (*Figure 1—figure supplement 3b*). Despite reduced overall accuracy, it is evident that DAPI and other DNA dyes may be an informative and cheap alternative to histone staining in at least some applications when the analyzed epigenetic landscape are very distinct.

Of note, the compound library used in this study included Pan HDACi, Class I HDACi, and Class I HDACi, known to also target HDAC6. HDAC inhibitors targeting both Class I and HDAC six displayed the same profile as Pan HDAC, and DA showed the two categories to be undistinguishable. This was likely due to the high expression of HDAC Class I and HDAC six and low expression of other HDACs in the GBM2 glioblastoma line (*Figure 1—figure supplement 4a,b,c*).

Of the 85 compounds tested, 7 (8.2%) were identified as active but were misclassified by MIEL. One of these was valproic acid, a commonly used anticonvulsant (*Peterson and Naunton, 2005*) which also functions as a Pan HDAC inhibitor at high concentrations (*Phiel et al., 2001*). Though valproic acid is expected to inhibit HDACs only at high concentrations (>1.2 mM), a short 24 hr treatment induced detectable epigenetic changes even at low concentrations (<30 µM). However, when we quantified H3K27ac and H3K27me3 immunofluorescence intensity at these concentrations, no increase in histone acetylation or decrease in histone methylation similar to other Pan HDAC inhibitors (TSA, SAHA; *Figure 1—figure supplement 5a*) was seen. To test, whether observed epigenetic changes resulted in corresponding transcriptomic alterations, we sequenced RNA from GBM2 cells treated with either DMSO, TSA, SAHA or valproic acid (15 µM) for 24 hr and identified all genes altered by at least one of the drugs (as compared to DMSO; 118 genes). This analysis indicated that the Pan HDAC inhibitors induced similar transcriptomic changes that were not apparent in the transcriptomic profile of valproic acid-treated cells (*Figure 1—figure supplement 5b*). To test whether MIEL profiles reflected global drug-induced transcriptomic profiles, FPKM values for all expressed genes (FPKM >1 in at least one cell population) were used to calculate the Euclidean distance between all 4 cell populations. FPKM-based distances were then correlated to image texture feature-based distances, which yielded a high and significant correlation between these metrics (R = 0.91, pv <0.05; *Figure 1—figure supplement 5c*).

Taken together, these demonstrate a unique ability of the MIEL approach to identify epigenetically active compounds, to accurately categorize them according to their molecular mechanism of action, and to detect off-target effects of compounds with known mechanism of action.

## Unbiased detection of compound concentration-dependent effect on cellular epigenetic state

As drugs vary in potency, predicting the function of unknown drugs relies on generating functional category-specific profiles that remain valid over a range of activity levels. To determine whether MIEL could correctly identify the functional category of drugs with different potencies, we treated GBM2 cells with drugs from several active categories at a range of concentrations (0.1, 0.3, 1, 3, 10 µM) and conducted DA aimed at separating the different concentrations in each class. We found that for most drug categories tested (inhibitors of: Aurora, JAK, SIRT and EZH1/2), DA yielded low

average classification accuracies (*Figure 2a* - Aurora kinase: 43.3%; *Figure 2—figure supplement 1a* - EZH1/2: 62.5%, SIRT:4 6.2%, JAK: 37.2), indicating similar MIEL profiles across tested drug concentrations. However, Pan HDAC and HDAC Class I inhibitors displayed progressive profile changes, allowing DA to separate the different concentrations at higher accuracy (*Figure 2a* – HDAC Pan: 80.9%; *Figure 2—figure supplement 1a* - HDAC Class I: 82.2%).

In addition to their on-target effect, drugs may induce epigenetic alterations through toxicity and stress. To estimate the impact of toxicity on drug induced profile changes and its contribution to drug misclassification across a range of concentrations, we plotted z-scored distance from DMSO (effect size) against z-scored nuclei count (a proxy for cytotoxicity) for GBM2 cells treated at a range of drug concentrations (0.1, 0.3, 1, 3, 10 μM). This demonstrated that some compound classes, such as Aurora and JAK inhibitors, induce epigenetic alterations only in concentrations at which cell count is significantly reduced, whether through toxicity or direct effect on proliferation (*Figure 2b* – dark blue and pink respectively), while other compounds, such as HDAC inhibitors, characteristically have a concentration range where epigenetic alterations are not accompanied by reduced cell counts (*Figure 2b* – green and yellow). Interestingly, both SIRT and EZH1/2 (*Figure 2b* – light-blue and red, respectively) inhibitors affected significant epigenetic changes without inducing significant changes in cell count.

These results indicated the MIEL platform is ideally positioned to analyze dose-dependent effects from drug treatment. In particular, our data suggest that low (0.1 μM) and high (10 μM) concentration of HDAC inhibitors resulted in distinct and separable epigenetic landscapes, suggesting potentially distinct chromatin/gene expression profiles and divergent biological outcomes when using a low vs high concentration of such compounds.

## MIEL profiles are coherent across multiple cell lines

Testing candidate drugs in multiple cell lines can help gauge their inclusivity and identify tumor subtypes that do not respond to a specific drug or drug class. To test whether MIEL readouts were coherent across multiple glioblastoma TPCs, we treated 4 cell lines with a subset of drugs from the epigenetic library (57 drugs), derived phenotypic profiles, and calculated their effect size (z-scored Euclidean distance from DMSO replicates). This revealed a significant positive correlation between all 4 cell lines pointing to similarities in their drug sensitivity profiles and demonstrating the robustness of the MIEL read out (*Figure 2c,d*). To assess the ability of MIEL to group compounds by function across multiple cell lines we employed DA to classify DMSO and drug treated TPCs across these 4 GBM lines. This analysis enabled accurate separation of cells treated with drugs modulating distinct functions, such as EZH1/2 or SIRT inhibitors (5 and 3 compounds respectively; mean 100% accuracy; *Figure 2e*). However, we were unable to separate drug subclasses with similar functions, such as class I and pan HDACs inhibitors (6 and 17 compounds respectively; mean accuracy 76.8%; *Figure 2e*). These results demonstrate the ability of MIEL to correctly categorize by function drugs with varying degrees of potency across multiple cells lines.

Finally, although individual drug activity correlated well across cells lines, the magnitude of the effect for some drug classes was highly correlated to the expression levels of the target gene. For example, SIRT inhibition was significantly more effective in lines showing reduced Sirt1 expression (the main SIRT to deacetylate histone 3; n = 4 compounds, p<0.02; *Figure 2—figure supplement 1b,c*), and there was a significant inverse correlation between Sirt1 expression and the effect size (R = −0.87; *Figure 2—figure supplement 1c*). These results further highlight the sensitivity of MIEL and its ability to reflect internal transcriptomic differences between cell populations.

## MIEL ranks compounds with similar function by activity

MIEL analysis indicated that the magnitude of drug induced profile changes, as measured by distance from DMSO replicates, varies between individual drugs within each drug class (*Figure 3—figure supplement 1a*). To test whether these differences are biologically meaningful, we correlated MIEL-based activity readouts with the ability of epigenetic drugs to synergize with other treatments as these are often designed to work as part of a combination therapy (*Lee et al., 2017*; *Romani et al., 2018*). One common approach is to use epigenetic drugs to sensitize tumor cells to standard of care cytotoxic treatments (*Strauss and Figg, 2016*; *Zhou et al., 2015*; *Li et al., 2017*; *Entin-Meer et al., 2007*), such as radiation and temozolomide (TMZ), which are used to treat

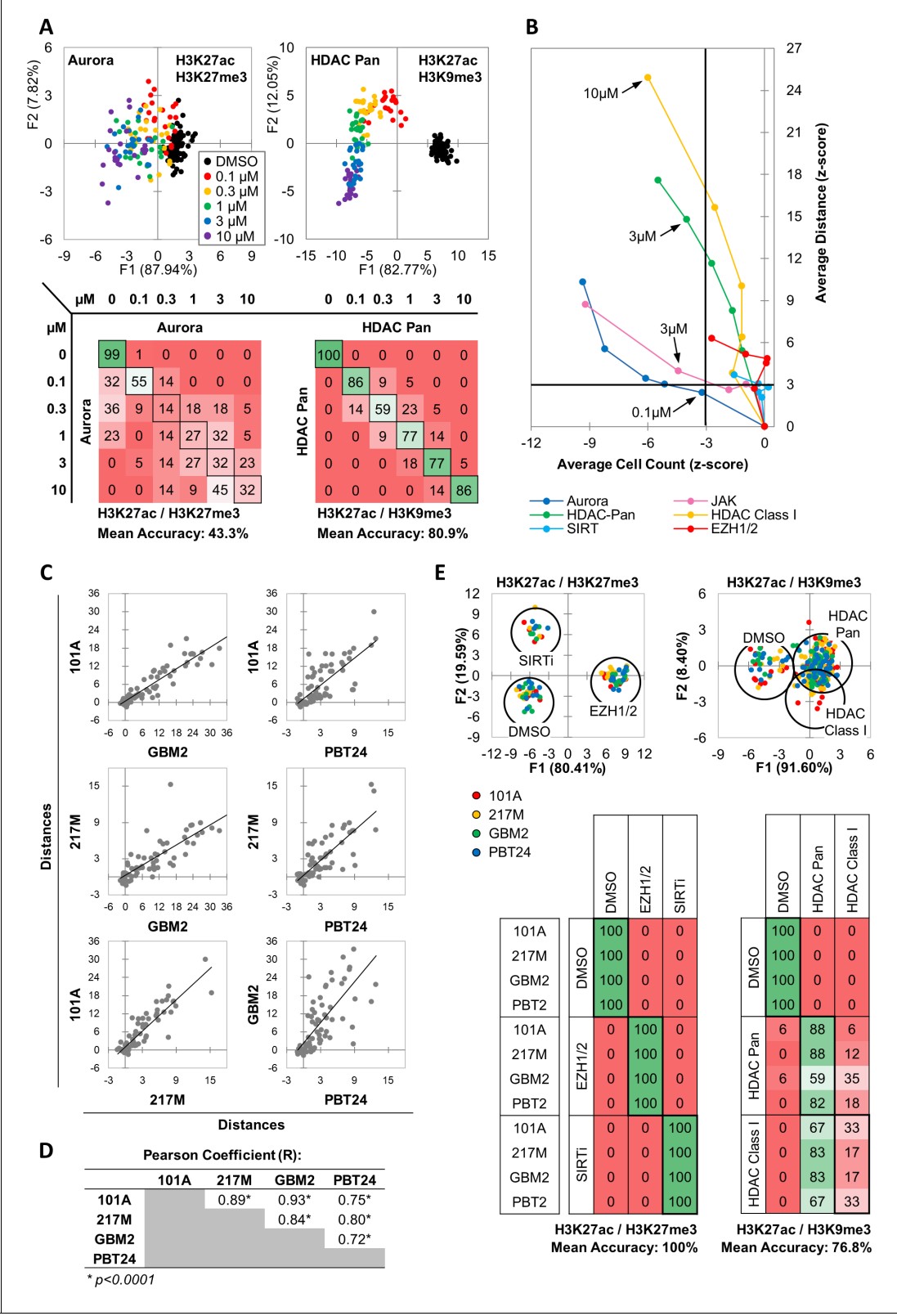

**Figure 2.** MIEL distinguishes between multiple categories of epigenetic drugs. (**A**) Quadratic discriminant analysis using texture features derived from images of GBM2 cells treated with DMSO, 0.1, 0.3, 1, 3 or 10 µM Aurora kinase (n = 11 compounds, two replicates) or HDAC Pan inhibitors (n = 11 compounds, two replicates) stained for either H3K9me3+H3k27ac or H3K27me3 + H3K27ac. Scatter plots depict the first two discriminant factors for each cell population (drug replicate). Confusion matrixes showing results for the discriminant analysis. Numbers represent the percent of replicates

*Figure 2 continued on next page*

Figure 2 continued

classified correctly (diagonal) and incorrectly (off the diagonal). (B) Scatter plot comparing the magnitude of effect (average z-scored Euclidean distances from DMSO) to drug-induced cytotoxicity (average z-scored cell count). Euclidean distance was calculated using image texture features derived from images of H3K27ac + H3K27me3 (Aurora, JAK, SIRT, EZH1/2) or H3K27ac + H3K9me3 (HDAC Pan, HDAC Class I). Distances and cell counts represent average of all compounds in each category; $n_{Aurora}$ = 11, $n_{EZH1/2}$=5, $n_{HDAC\ Class\ I}$=7, $n_{HDAC\ Pan}$=43, $n_{JAK}$ = 15, $n_{SIRTi}$ = 4). Arrows denote the lowest concentration at which compounds of each category induce significant cytotoxicity. (C) Scatter plots comparing the z-scored Euclidean distances from DMSO replicates across 4 GBM lines (n = 57 compounds, z-score for each compound is the average of 3 technical replicates). Euclidean distances were calculated using image texture features derived from images of H3K27ac and H3K27me3 or H3K27ac and H3K9me3. (D) A table summarizing the Pearson coefficient and statistical significance of z-scored Euclidean distances shown in 'C.' (E) Quadratic discriminant analysis using texture features derived from images of GBM2, PBT24, 101A, 217 M cells treated with either DMSO, 5 EZH1/2 inhibitors, 3 SIRT inhibitors, 6 Class I HDAC inhibitors or 17 Pan HDAC inhibitors. Features derived from images of cells stained for H3K27me3 + H3K27ac (EZH1/2, SIRT) or H3K27ac + H3K9me3 (HDACi). Scatter plots depicting the first two discriminant factors for each cell population (two replicates per drug per cell line) color coded according to cell line. Confusion matrix showing classification results for the discriminant analysis (test set, one replicate per drug per cell line). Numbers represent the percent of compounds classified correctly (diagonal) and incorrectly (off the diagonal).

The online version of this article includes the following figure supplement(s) for figure 2:

**Figure supplement 1.** MIEL distinguishes between multiple categories of epigenetic drugs across different drug concentrations.

glioblastoma. To identify drug classes that sensitize glioblastoma TPCs to cytotoxic therapy, GBM2 cells were treated with epigenetic drugs for 2 days prior to radiation or TMZ. Cytotoxic treatment was carried out for 4 days at levels that induced a 50% reduction in cell numbers (1Gy radiation or 200 µM TMZ; *Figure 3a*). At the end of day six treatment, cells were counted, and a combined drug index (CDI) was calculated (see Materials and methods). Though we did not identify any drugs that synergized (CDI < 0.7) with the radiation therapy (*Figure 3b*, right panel), multiple PARP and BET inhibitors (PARPi and BETi) sensitized cells to TMZ (*Figure 3b*, left panel).

PARPi have been extensively studied in this context and have been shown to function through multiple non-epigenetic mechanisms such as PARP trapping (*Murai et al., 2012*; *Lord and Ashworth, 2017*; *Kedar et al., 2012*). Consistent with this, most PARPi did not induce detectable epigenetic changes using MIEL (*Figure 3d*, *Figure 3—figure supplement 1b*), and we found no correlation between the magnitude of epigenetic changes as measured by MIEL and CDI (*Figure 3d* – bottom panel). To date, only a single report utilizing the BETi OTX015 (*Berenguer-Daizé et al., 2016*) has pointed to synergy with TMZ, prompting us to validate this finding in five additional glioblastoma lines. In three lines, BETi increased the TMZ effectiveness (average CDI: 454M 0.76 ± 0.28, PBT24 0.78 ± 0.12 and GBM2 0.51 ± 0.2; Mean ± SD; n = 11 BETi; *Figure 3c*). In the other three lines, the drugs did not synergize and, in many cases, were found to be protective against (CDI > 1) TMZ (average CDI: SK262 1.4 ± 0.26, 101A 1.4 ± 0.22 and 217M 1.2 ± 0.21; Mean ± SD; n = 11 BETi; *Figure 3c*; p- values for all pairwise comparisons *Figure 3c*).

We detected only few BETi-induced epigenetic changes in our initial screen conducted over 24 hr (*Figure 1b*). However, following a 6 days treatment 6 out of 11 BETi induced significant (average z-scored distance from DMSO replicates >3) epigenetic changes in all cell line tested (*Figure 3d*, *Figure 3—figure supplement 1b*). In lines displaying TMZ and BETi synergy, the degree of BETi activity, as measured by MIEL, significantly correlated with the degree of synergism (*Figure 3d* – top panel). This demonstrated that for individual compounds, MIEL can predict relative drug activity and suggests an epigenetic component for the mechanism of BETi-TMZ synergy.

## BET inhibitors decrease expression of MGMT

$O^6$-alkylguanine DNA alkyltransferase (MGMT), which provides the main line of defense against DNA alkylating agents such as TMZ, has been found to be epigenetically silenced through DNA methylation in a large fraction of glioblastoma tumors (*Karayan-Tapon et al., 2010*; *Hegi et al., 2005*). To gain a better understanding of the mechanism by which BETi sensitize glioblastoma TPCs to TMZ treatment, we quantified MGMT expression in the six lines tested using qPCR. This analysis showed that while all lines expressed similar BET transcription factors (TFs) levels, such as Brd2 (*Figure 3e*), and were thus susceptible to BET inhibitors, only the three lines displaying BETi-TMZ synergy expressed MGMT (*Figure 3e*). Treating those three lines with BETi, dramatically reduced MGMT expression (*Figure 3f*). Finally, combining BET inhibitors with the MGMT inhibitor

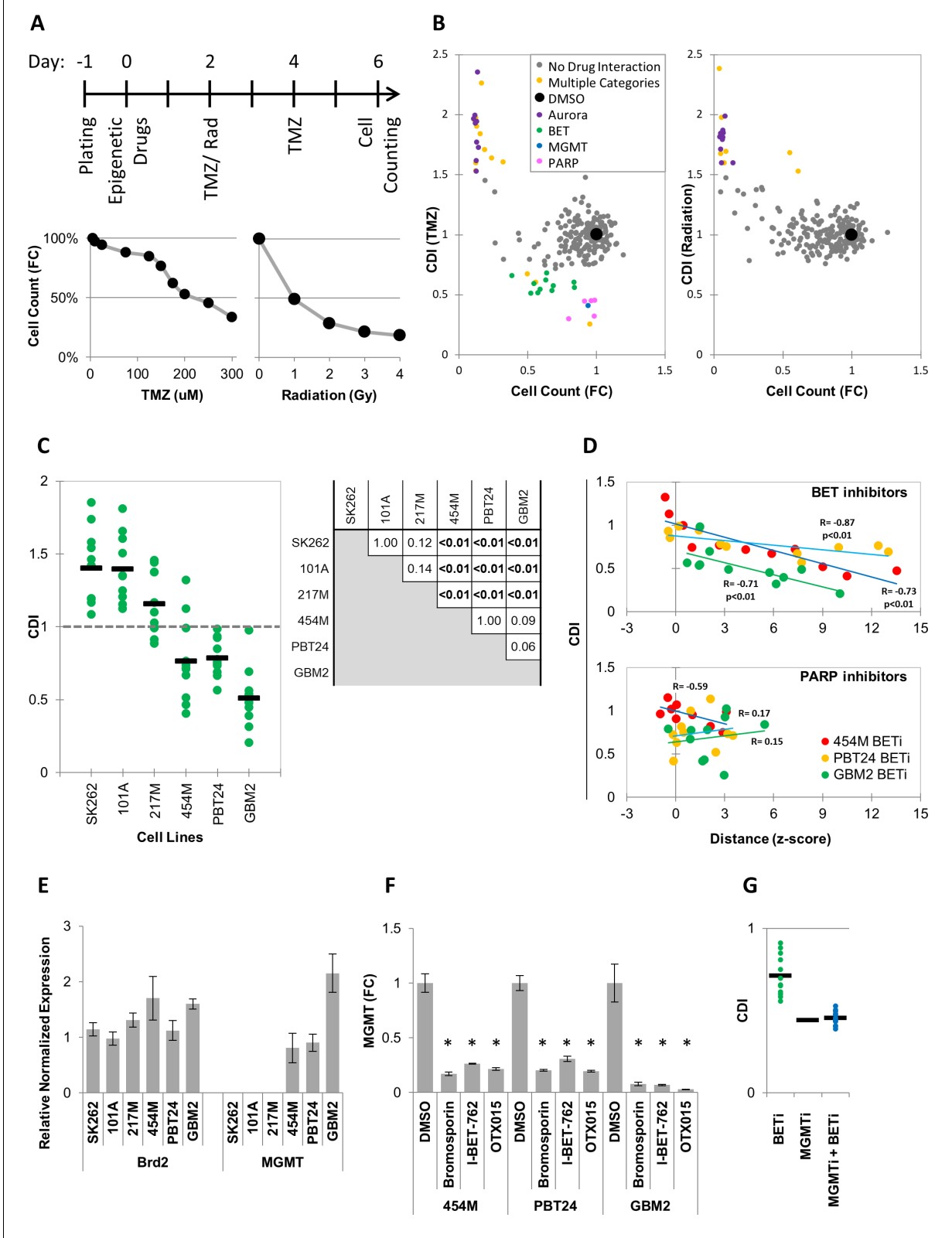

**Figure 3.** MIEL can be used to rank candidate drugs by activity. (**A**) Top: Scheme describing the experimental setup used to identify synergy between epigenetic drugs and radiation or TMZ. Bottom: Scatter plots showing the fold reduction in GBM2 cell count following a 4 day treatment with varying TMZ concentration and radiation doses. (**B**) Scatter plots showing fold change in cell count (compared to DMSO treated cells) and coefficient of drug interaction (CDI) for synergy with TMZ (left) and radiation (right) for each drug (n = 222, values represent the average of 3 technical replicates). (**C**)

*Figure 3 continued on next page*

*Figure 3 continued*

Graph showing individual and average CDI for BET inhibitors in 6 GBM lines (n = 11 drugs, average of 3 technical replicates; p-values calculated by ANOVA using Tukey's HSD for multiple comparisons between lines and shown in table). (D) Scatter plot showing the correlation between CDI and MIEL-derived activity (z-scored Euclidean distance from DMSO) of BET and PARP inhibitors ($n_{BETi}$ = 11; $n_{PARPi}$ = 10; values represent the average of 3 technical replicates) in 3 GBM lines (454M, PBT24, GBM2). (E) Bar graph showing the relative normalized expression of Brd2 and MGMT in 6 GBM lines as measured by qPCR (Mean ± SD; n = 3 technical repeats). (F) Bar graph showing fold reduction in MGMT expression following treatment with BET inhibitors in three different GBM lines as measured by qPCR (Mean ± SD; n = 3 technical repeats). (G) Graph showing individual and average TMZ sensitization CDI for BETi, MGMTi (Lomeguatrib) and BETi and MGMTi in GBM2 cells (n = 11 drugs, values represent the average of 3 technical replicates).

The online version of this article includes the following figure supplement(s) for figure 3:

**Figure supplement 1.** MIEL can report relative drugs activity.

Lomeguatrib did not increase sensitivity to TMZ above the levels conferred by Lomeguatrib alone (*Figure 3g*).

In sum, we have discovered that several BETi synergized with TMZ treatment by reducing MGMT expression. We applied MIEL to rank BETi according to their magnitude of epigenetic effect and demonstrated that they ranks according their ability to synergize with TMZ suggesting that their mechanism of action involves epigenetic change. In contrast, the activity of PARP inhibitors didn't correlate with magnitude of epigenetic effect, suggesting an alternative mechanism of action. Thus, we propose that the MIEL approach is well positioned to systematically analyze and identify epigenetically active compounds, then provide critical initial information for their mechanism of action.

## MIEL discriminates between multiple cell fates

By altering histone and DNA modifications, epigenetic drugs have a direct effect on the MIEL read-out. To test the ability of MIEL to identify and classify in-direct epigenetic changes we tested its utility for identifying drugs inducing GBM differentiation. Previous attempts to design screening strategies for this purpose have met with multiple difficulties. One critical problem is the lack of informative markers faithfully reporting GBM differentiation that could be used for high-throughput screening (*Patel et al., 2014*). The lack of informative markers for GBM differentiation and the ability of MIEL to identify compounds producing desired epigenetic alterations prompted us to test the feasibility of using this approach to screen for drugs inducing GBM TPCs differentiation.

For this, we first tested the ability of MIEL to discriminate between different cell fates. We analyzed 3 cell types: primary human fibroblasts, induced pluripotent stem cells (iPSCs) derived from these fibroblasts, and neural progenitor cells (NPCs) differentiated from the iPSCs. The fibroblasts were isolated from three unrelated donors (WT-61, WT-101, WT-126) and used to obtain corresponding iPSC and NPC lines. Cellular identities of the 3 cell types were verified by immune-fluorescence for Sox2 and Oct4 (*Figure 4a*), and MIEL analysis was carried out using data from either H3K4me1 and H3K9me3 or H3K27ac and H3K27me3 staining, with both combinations providing similar results. Multivariate centroids were calculated for each cell population and plotted on a distance map to visualize the relative Euclidean distance between various cell populations. The fibroblasts, iPSCs, and NPCs each segregate to form three visually distinct territories (*Figure 4—figure supplement 1c*). We separated the nine lines by cell-fates using DA, which showed an accurate separation of the different cell-fates across all three donors (average accuracy 100%; *Figure 4b*, *Figure 4—figure supplement 1e*). A similar analysis aimed at separate the different donors showed only low accuracy (average accuracy 55.5%; *Figure 4c*, *Figure 4—figure supplement 1f*). To determine whether it was possible to discriminate between individual cells with different fates, a Support Vector Machine (SVM) classifier was trained on a subset of fibroblasts, iPSCs, and NPCS from the three donors. Classification of the test set indicated a high degree of separation between the different fates at a single cell level (*Figure 4—figure supplement 1b,d*). Additionally, MIEL analysis (using only H3K9me3) was able to discriminate between primary hematopoietic cell types freshly isolated from mouse bone marrow, namely lymphoid, myeloid, and stem/progenitors (*Figure 4—figure supplement 2*). However, closely related hematopoietic stem and progenitor cells were not readily separated (*Figure 4—figure supplement 2*).

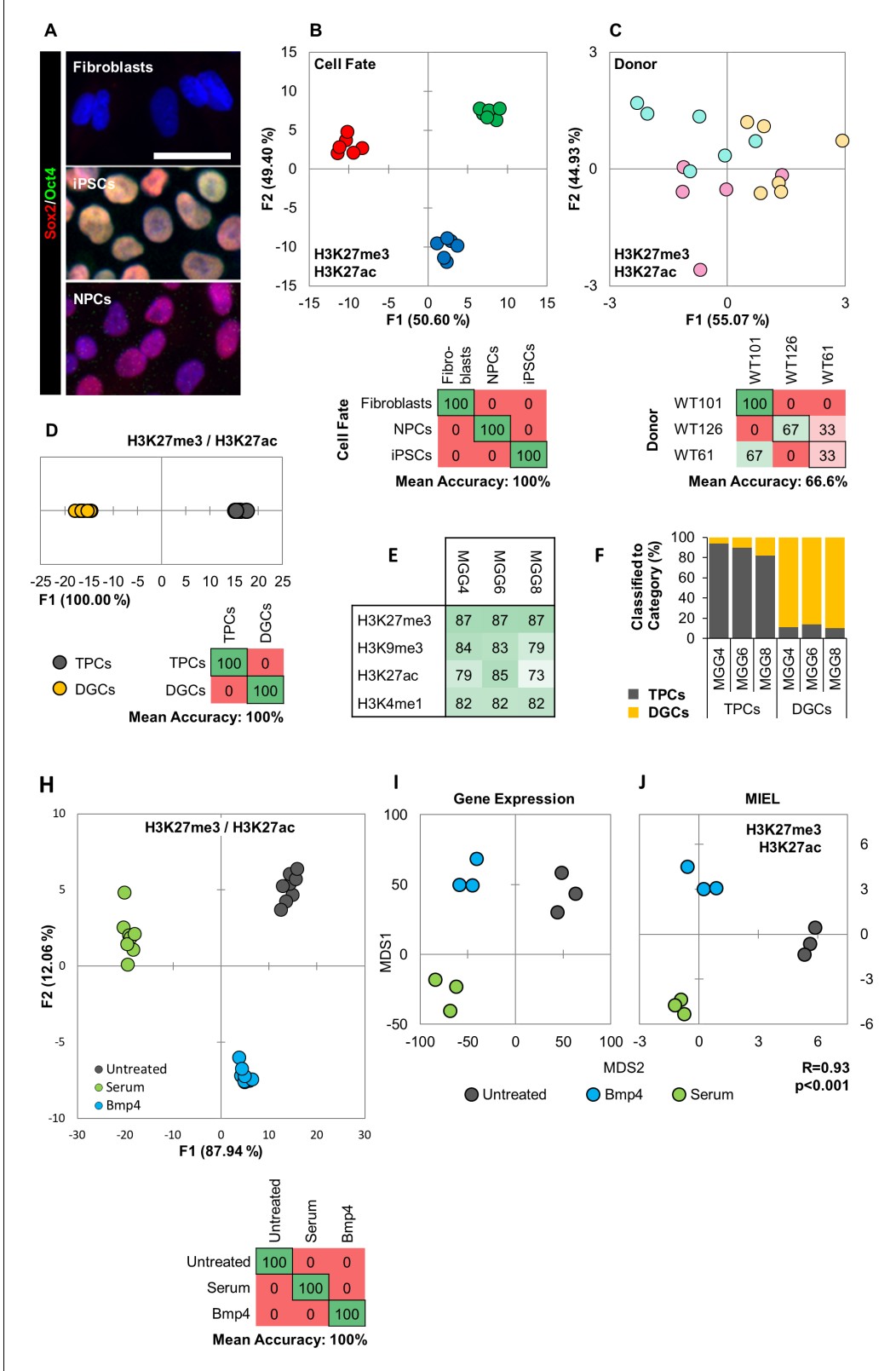

**Figure 4.** MIEL can distinguish between cell fates and identify glioblastoma differentiation. (**A**) Hoechst 33342 stained (blue), and Sox2 (red) and Oct4 (green) immunofluorescence labeled fibroblasts (Sox2⁻/Oct4⁻), iPSCs (Sox2⁺/Oct4⁺) and NPCs (Sox2⁺/Oct4⁻). Scale bar, 50 μm. (**B, C**) Quadratic discriminant analysis separating either cell fates or cell lines using texture features derived from images of fibroblasts, iPSCs, and NPCs lines from three human donors (WT-61, WT-101 and WT-126; three technical replicates each); stained for H3K9me3 and H3K4me1. (**B**) Discriminant analysis separating

*Figure 4 continued on next page*

Figure 4 continued

the different cell types. Scatter plot depicting the first two discriminant factors for each cell population (two replicate per cell line and cell type). Confusion matrixes showing classification results for discriminant analysis (test set: one replicate per cell line and cell type) Numbers represent the percent of correctly (diagonal) and incorrectly (off the diagonal) classified cell populations. (C) Discriminant analysis attempting to separate different cell lines. Scatter plot depicting the first two discriminant factors for each cell population (two replicates per cell line and cell type). Confusion matrixes showing classification results for discriminant analysis (test set: one replicate per cell line and cell type). Numbers represent the percent of correctly (diagonal) and incorrectly (off the diagonal) classified cell populations. (D, E, F) TPC and DGC cell lines derived simultaneously from tumors of 3 human donors (MGG4, MGG6, MGG8; three technical replicates each); stained for H3K9me3, H3K4me1. (D) Quadratic discriminant analysis separating TPCs and DGCs using image texture features. Scatter plot depicting the first discriminant factor for each cell population (two replicates per cell line). Confusion matrix showing classification results for discriminant analysis (test set: one replicate per cell line). Numbers represent the percent of correctly (diagonal) and incorrectly (off the diagonal) classified cell populations. (E) Pairwise classification of single TPC and DGC cells using an SVM classifier trained on texture features derived from images of H3K27me3, H3K9me3, H3K27ac, or H3K4me1. Numbers correspond to the percent of correctly classified cells for each line using indicated epigenetic marks. (F) Bar graph showing results of SVM classification for single TPC and DGC cells using a classifier trained on texture features derived from images of H3K27ac and H3K27me3 marks in the MGG4 line. (H) Quadratic discriminant analysis using texture features derived from images of untreated or 2 days serum or Bmp4 treated GBM2, 101A, SK262 and 454 M cells (three replicates per cell lines per treatment) and stained for H3K9me3, H3K4me1. Scatter plot depicting the first two discriminant factors for each cell population (two replicates per cell lines per treatment). Confusion matrix showing classification results for discriminant analysis (test set: one replicate per cell line per treatment). Numbers represent the percent of correctly (diagonal) and incorrectly (off the diagonal) classified cell populations. (I) Distance map depicting the relative Euclidean distance between the transcriptomic profiles of DMSO-, Bmp4- and serum-treated GBM2 cells calculated using FPKM values of all expressed genes (14,376 genes; FPKM > 1 in at least one sample). Each treatment in triplicates. (J) Distance map depicting the relative Euclidean distance between the multiparametric centroids of DMSO-, Bmp4- and serum-treated GBM2 cells calculated using texture features derived from images of H3K27ac and H3K27me3 marks. Each treatment in triplicates. R denotes Pearson correlation coefficient.

The online version of this article includes the following figure supplement(s) for figure 4:

Figure supplement 1. MIEL can distinguish between multiple cell fates.
Figure supplement 2. MIEL can distinguish between cells from different hematopoietic lineages.
Figure supplement 3. Serum and Bmp4 reduce expression of genes associated with undifferentiated glioblastoma TPCs.
Figure supplement 4. GO analysis of transcriptomic changes induced by serum and Bmp4.
Figure supplement 5. Serum and Bmp4 treatments induce distinct epigenetic and transcriptomic changes.

These results underscore MIEL's ability to discriminate multiple different cell types and differentiation states uniquely based on their single-cell epigenetic landscapes both in cultured and primary cells of human and mouse origin.

## MIEL determines the signatures of glioblastoma stem cells and differentiated glioblastoma

We tested MIEL's ability to distinguish TPCs and differentiated glioma cells (DGCs), derived from the same primary human GBMs (*Suvà et al., 2014*). Three TPC/DGC pairs were derived in parallel from three genetically distinct glioblastoma tumor samples (MGG4, MGG6, and MGG8) over a 3 month period using either serum-free FGF/EGF conditions for TPCs or 10% serum for DGCs (*Suvà et al., 2014*). Visualization using distance maps demonstrated that TPCs and DGCs segregate to form two visually distinct territories (*Figure 4—figure supplement 1g*) and were separated with high accuracy using DA (mean accuracy 100%; *Figure 4d*). SVM-based pairwise classification of single cells distinguished TPCs from their corresponding DGC lines with an average accuracy of 83%, using any of the four epigenetic marks tested (H3K27me3, H3K9me3, H3K27ac, and H3K4me1; *Figure 4e*). An SVM classifier derived from images of the MGG4 TPC/DGC pair separated all 3 TPC/DGC pairs with 88% average accuracy, providing proof of principle for the derivation of a signature for non-tumorigenic cells obtained following serum differentiation of primary glioblastoma cells (*Figure 4f*).

These findings suggest that MIEL can readily distinguish undifferentiated TPCs from differentiated DGCs based on multiparametric signatures of these glioblastoma cells using only the patterns of universal epigenetic marks.

## Short-term treatment with serum or Bmp4 initiates TPC differentiation

For the purpose of establishing a screening protocol, we tested whether short serum or Bmp4 treatment is sufficient to induce a differentiation-like phenotype in TPCs. We treated several glioblastoma cell lines for 3 days with either serum or Bmp4, then quantified expression of core transcription

factors previously shown to determine the TPC transcriptomic program of TPCs (*Suvà et al., 2014*). Immunostaining revealed that the four transcription factors Sox2, Sall2, Brn2 and Olig2 were down-regulated by both serum and Bmp4 in a cell line-dependent manner (*Figure 4—figure supplement 3a*). RNAseq analysis of serum- and Bmp4-treated GBM2 cells revealed that 3 days of treatment reduced (vs untreated cells) expression of most genes previously found to constitute the transcriptomic stemness signature (*Patel et al., 2014*) (*Figure 4—figure supplement 3b*). Additionally, both serum and Bmp4 were found to attenuate TCP growth rate (*Figure 4—figure supplement 3c*). To identify the cellular processes altered by these treatments, we conducted differential expression analysis. Expression of 4852 genes was significantly altered (p<0.01 and −1.5 < Fold Change>1.5) by either serum or Bmp4. Gene Ontology (GO) analysis of these altered genes indicated enrichment in multiple GO categories consistent with initiation of TPC differentiation; these include cell cycle, cellular morphogenesis associated with differentiation, differentiation in neuronal lineages, histone modification, and chromatin organization (*Figure 4—figure supplement 4*).

These results demonstrate that a 3 day treatment with either serum or Bmp4 is sufficient to induce transcriptomic changes characteristic of TPC differentiation. Previous work indicated distinct features of glioblastoma differentiation induced with BMP compared to serum (*Carén et al., 2015*). Indeed, we observed distinct expression changes, including differences in expression of genes regulating chromatin organization and histone modifications (*Figure 4—figure supplement 5a, b*), between serum- and Bmp4-induced glioblastoma differentiation.

## MIEL detects epigenetic changes following short-term serum or Bmp4 treatment

To test the ability of MIEL to detect short term TPCs differentiation we treated four genetically distinct glioblastoma lines with serum or BMP4, then conducted MIEL analysis using either H3K9me3 and H3K4me1 or H3K27ac and H3K27me3. Discriminant analysis allowed high accuracy separation of these treatments across all cell lines using both histone modification combinations (mean accuracy 100%; *Figure 4h*; *Figure 4—figure supplement 5c*).

The global gene expression profile represents a gold standard for defining the cellular state (*Liang et al., 2005*). Therefore, we correlated the relative distances between distinct cellular states, using MIEL-based and global gene expression-based metrics. We sequenced untreated and 3 days serum- or Bmp4-treated GBM2 TPCs (three replicates each) and used FPKM values of all expressed genes (FPKM >1 in at least one cell population) to calculate the Euclidean distance matrix between all cell populations. FPKM-based distances were then correlated to image texture feature-based distances. The resulting Pearson correlation coefficient of R = 0.93 (p<0.001) suggests a high correlation between these two metrics (*Figure 4i,j*), demonstrating that MIEL is capable of distinguishing closely related glioblastoma differentiation routes induced by serum or BMP and validating the robustness of the MIEL approach for analyzing glioblastoma differentiation.

## MIEL based screen for compounds inducing TPCs differentiation

To test whether MIEL can identify compounds inducing GBM TPCs differentiation based on serum/Bmp4 signature, we screened the Prestwick compound library (1200 compounds). GBM2 TPCs were treated for 3 days with Prestwick compounds at 3 μM fixed, then immune-labeled for H3K27ac and H3K27me3. GBM2 cells treated with DMSO, serum, BMP4, or compound were compared within the same plate (to avoid imaging artifacts and normalization issues).

To identify epigenetically active compounds, we calculated the Euclidean distance to the DMSO center for each DMSO replicate and Prestwick compound. Distances were z-scored, and active compounds were defined as compounds for which z-scored distance was greater than 3. Compounds with less than 50 cells imaged were considered toxic and excluded from analysis. This analysis detected 144 active compounds. To identify compounds inducing epigenetic changes reminiscent of serum- BMP4-induced differentiation, we used quadratic DA to build a model separating untreated, serum- and Bmp4-treated cells and classified all 144 active compounds to these categories (*Figure 5a,b*). A total 31 compounds were classified as similar to either serum or Bmp4 (i.e., differentiated). Of these, 20 compounds belonged to 1 of the following four categories: Na/K-ATPase inhibitors of the digoxin family, molecules that disrupt microtubule formation or stability, topoisomerase inhibitors, or nucleotide analogues that disrupt DNA synthesis (*Figure 5b*). To further narrow down

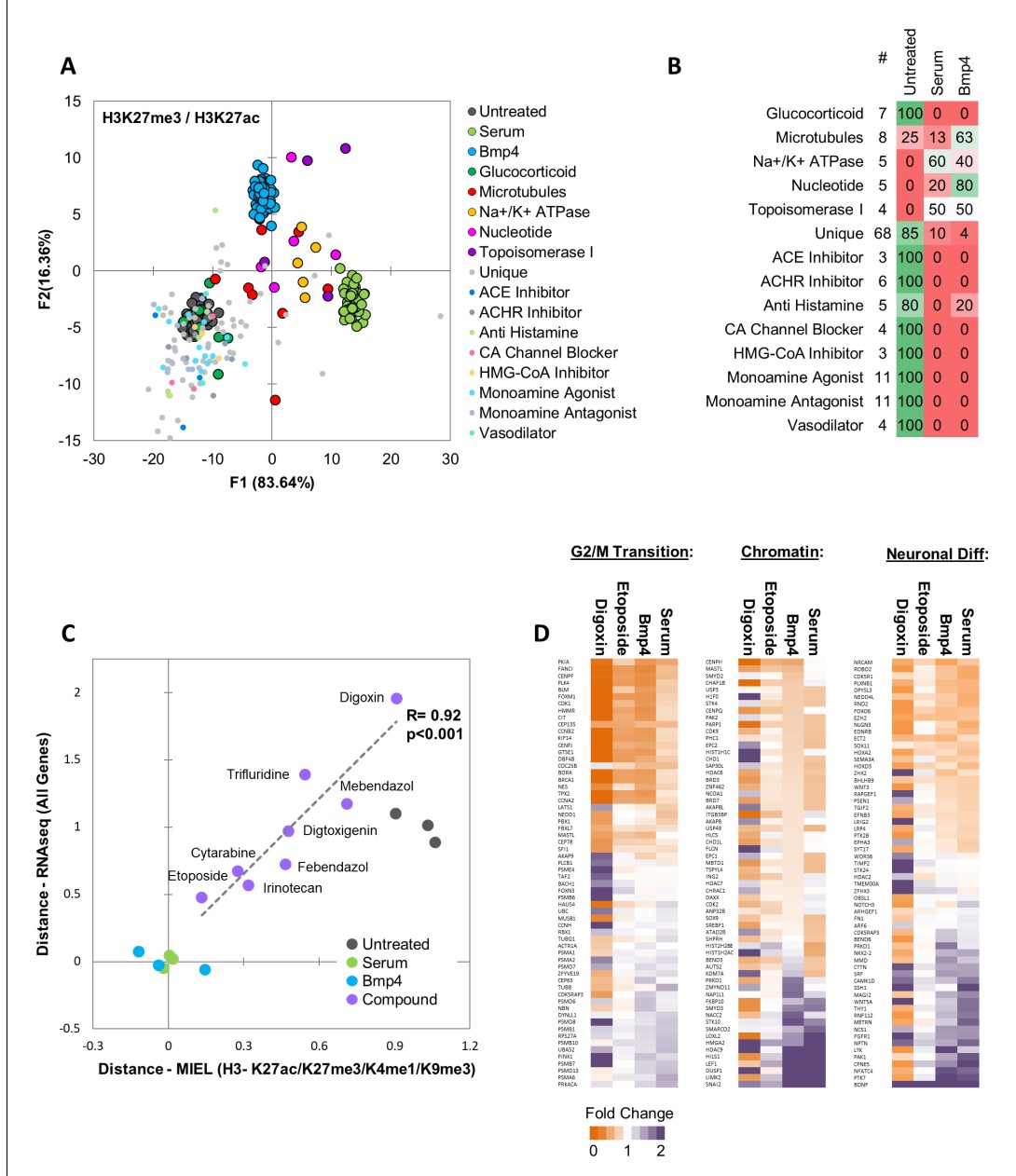

**Figure 5.** MIEL prioritizes small molecules based on serum/Bmp4 differentiation signature. Quadratic discriminant analysis using texture features derived from images of untreated, serum-, Bmp4- and compound-treated GBM2 cells stained for H3K27me3, H3K27ac. Model was built to separate untreated, serum- and Bmp4-treated cells (60 technical replicates each). (A) Scatter plot depicting the first two discriminant factors for each population. (B) Confusion matrix showing classification of epigenetically active Prestwick compounds. Numbers depict the percent of compounds from each category classified as either untreated, serum or Bmp4 treated. (C) Scatter plot showing the correlation of gene expression profile-based ranking and MIEL-based ranking for eight candidate drugs, untreated, serum- or Bmp4-treated GBM2 cells. Euclidean distance to serum- or Bmp4-treated GBM2 cells was calculated using transcriptomic profiles (vertical axis) or texture features derived from images of H3K27ac and H3K27me3, H3K9me3, and H3K4me1 marks (horizontal axis). Distances were normalized to untreated and serum- or Bmp4-treated GBM2 cells. (D) Heat maps showing fold change in expression of select genes taken from the Gene Ontology (GO) list: cell cycle G2/M phase transition (GO:0044839), chromatin modification (GO:0006325), and regulation of neuron differentiation (GO:0045664). R denotes Pearson correlation coefficient. Drug concentrations a-c: febendazole = 0.5 µM, mebendazole = 0.5 µM, cytarabine = 0.3 µM, trifluridine = 3 µM, irinotecan = 0.5 µM, etoposide = 0.3 µM, digitoxigenin = 0.3 µM, digoxin = 0.3 µM.

The online version of this article includes the following figure supplement(s) for figure 5:

**Figure supplement 1.** Functional categories of drugs prioritized by MIEL from the Prestwick library.

*Figure 5 continued on next page*

Figure 5 continued

**Figure supplement 2.** Drugs identified in screen lower proliferation rate of Glioblastoma cells but do not induce down regulation of key transcription factors.

the list of candidates, we conducted pairwise SVM classification of DMSO- and either serum- or BMP4-treated cells, then selected compounds that induced at least 50% of the cells to be classified as either serum- or BMP4-treated. We then calculated the Euclidean distance between candidate compounds and serum- or BMP4-treated cells and selected compounds where the distance to one or both treatments was less than the distance between DMSO and that treatment. This yielded 20 candidate compounds, of which 15 belonged to 1 of the four categories mentioned above; the top two compounds from each category were chosen for further analysis (*Figure 5—figure supplement 1a*).

GBM2 cells were treated for 3 days with DMSO, serum, Bmp4 or candidate compounds at 0.3,1 or 3 μM, fixed, and then immunostained for H3K27ac and H3K27me3. Using pairwise SVM based classifications of untreated cells and either serum- or Bmp4-treated cells we identified for each of the eight compounds the lowest concentration at which 50% or more of the cells were classified as treated (*Figure 5—figure supplement 1b*). These concentrations were used for all subsequent experiments (*Supplementary file 1* - Table S7). Because most of these compounds are known for their cytotoxic effects, we verified the growth rates of drug-treated glioblastoma cells. With the exception of digoxin, which was cytostatic, drug treatment resulted in growth rates comparable with that induced by serum or BMP4 (*Figure 5—figure supplement 2a*). We used immunofluorescence to test for expression of core TPC transcription factors (Sox2, Sall2, Brn2 and Olig2). Except for tri-fluridine, all compounds induced statistically significant reductions in Sox2; digoxin and digitoxigenin also induced a significant reduction of Sall2 and Brn2; Olig2 expression was unaltered by any treatment (*Figure 5—figure supplement 2b*).

Next, we investigated whether the compounds identified using MIEL can induce transcriptomic changes similar to serum and Bmp4 treatment and quantified the ability of MIEL to predict compounds best at mimicking these treatments. GBM2 cells were treated with DMSO, serum, Bmp4, or each of the eight candidate compounds; after 3 days, RNA was extracted and sequenced. Transcriptomic profiles of the eight compounds were ranked according to average Euclidean distance (based on FPKM values for all expressed genes) from serum and BMP4-treated cells. To safeguard against potential artefacts of cytotoxicity, we compared gene expression-based ranking with measured cellular growth rates from drug treatments and found no positive correlation (*Figure 5—figure supplement 2c*). Next, we compared Sox2 levels under all treatment conditions to determine whether expression of this transcription factor can identify drugs that best mimic serum or BMP4. We found no positive correlation between Sox2 expression and the transcriptomic-based rankings (*Figure 5—figure supplement 2d*), suggesting that Sox2 levels alone are insufficient to stratify the compounds. Finally, to compare MIEL-based signatures to the transcriptomic profile, we ranked MIEL readouts of cells treated with the eight drugs according to average Euclidean distance from serum- or Bmp4-treated cells (calculated using texture features derived from images of H3K27ac, H3K27me3, H3K9me3 and H3K4me1). Comparison of the MIEL-based metric with the gene expression-based metric revealed a high degree of positive correlation between MIEL- and gene expression-based rankings (Pearson correlation coefficient R = 0.92, p<0.001; *Figure 5c*). To further visualize these results, we constructed heat maps depicting fold change in expression levels of genes associated with several GO terms enriched by serum and Bmp4. Our top candidate, etoposide, altered expression of a large portion of genes in similar fashion to that of serum and BMP4; in contrast, the lowest-ranking candidate, digoxin, induced changes in gene expression, which were rather different from serum and BMP4 (*Figure 5d*).

Taken together, the above results reflect the unique ability of MIEL to identify molecules that shift epigenetic signature of glioblastoma TPCs towards DGCs. Critically, MIEL is capable of ranking such molecules according to their change-inducing potency and that ranking robustly correlate with global expression-based readouts of glioblastoma differentiation.

## Discussion

Here we have introduced MIEL, a novel method that expands phenotypic profiling to take advantage of universal biomarkers present in all eukaryotic cells by exploiting the patterns of chromatin organization and histone modification patterns. The pipeline we developed employs information derived from immunofluorescence images of specific histone modifications and is geared towards drug discovery and high-content screening. Focusing on compounds that modulate epigenetic writers, erasers, and readers, we have shown that MIEL markedly improves detection compared to conventional intensity-based thresholding approaches and enables their function based categorization. We have demonstrated that MIEL readouts are coherent across multiple compound concentrations and cell lines and can provide information regarding drug activity levels and their mechanism of action. We have also documented MIEL ability to robustly report cellular fate and provide proof of concept for identifying and prioritizing drugs inducing differentiation of glioblastoma TPCs.

### MIEL distinguishes between drugs classes with similar function

Previous studies have demonstrated that image-based profiling can distinguish between classes of compounds with very distinct functions, such as Aurora and HADC inhibitors (*Kang et al., 2016*). One objective of our study was to estimate the resolution of separation between categories of compounds with similar functions. We found that a single histone modification was sufficient to separate highly distinct classes (*Figure 1—figure supplement 3b*). However, separating similar classes (e.g., Aurora and JAK inhibitors, which affect histone phosphorylation, or Pan and Class I HADCs, which affect histone acetylation) required staining for at least one additional histone modification (*Figure 1d*). Despite their many advantages, cellular assays, including high-content assays, are often used as secondary screens for epigenetic drugs due to multiplicity of enzyme family members and an inability to determine direct enzymatic activity (*Martinez and Simeonov, 2015*). Consequently, MIEL's ability to separate closely related functional categories on top of other advantages make this profiling approach an attractive alternative for primary screens.

### Coherence across cell lines can provide vital input for personalized medicine

Phenotypic profiling methods have been previously used to identify genotype-specific drug responses by comparing profiles across multiple isogenic lines (*Breinig et al., 2015*). Here we show that activity of biologics (i.e., serum and Bmp4) that induces glioblastoma differentiation, as well as that of 57 epigenetic compounds, was significantly correlated across four different primary glioblastoma lines (*Figure 2c,d,e*; *Figure 4h*). We also showed that variation in activity levels correlated with target expression levels and that various categories can be distinguished across cell lines. Together, these suggest that MIEL could be used to identify cell lines showing an aberrant reaction to selected drugs and, therefore, aid in identifying optimal treatments for individual patients. Similar applications have previously been used to tailor specific kinase inhibitors to patients with chronic lymphocytic leukemia (CLL) who display venetoclax resistance (*Oppermann et al., 2016*).

### Non-cytotoxic treatments for Glioblastoma

Given the limited success of cytotoxic drugs in treating glioblastoma, we focused on alternative approaches: (1) epigenetic drugs aimed at sensitizing glioblastoma TPCs to such treatments, and (2) inducing glioblastoma differentiation. We have demonstrated MIEL's ability to rank candidate drug activity to correctly predict the best candidates for achieving the desired effect. The importance of this is highlighted in large (hundreds of thousands of compounds) chemical library screens, which typically identify many possible hits needing to be reduced and confirmed in secondary screens (*Hughes et al., 2011*; *Strovel et al., 2004*).

### BET inhibitors modulate expression of MGMT

Our results show a significant correlation between BET inhibitor activity, as defined by MIEL (*Figure 3d*), and their ability to synergize and increase TPC sensitivity to TMZ and reveal a previously unknown role for BET inhibitors in reducing MGMT expression (*Figure 3e,f,g*). Previous studies have demonstrated upregulation of several BET transcription factors in glioblastomas (*Pastori et al., 2014*; *Wadhwa and Nicolaides, 2016*) and multiple pre-clinical studies have investigated the

potential of BET inhibition as a single drug treatment for glioblastoma (*Xu et al., 2018*; *Ishida et al., 2017*; *Cheng et al., 2013*). However, while clinical trials with the BET inhibitor OTX015 demonstrated low toxicity at doses achieving biologically active levels, no detectable clinical benefits were found (*Hottinger et al., 2016*). This prompted approaches using drug combination treatments (*Ramadoss and Mahadevan, 2018*) such as combining HDACi and BETi (*Heinemann et al., 2015*; *Bhadury et al., 2014*). The mechanism by which BETi induces increased TMZ sensitivity has not been described. Recently, a distal enhancer regulating MGMT expression was identified (*Chen et al., 2018*). Activation of this enhancer by targeting a Cas9-p300 fusion to its genomic locus increased MGMT expression while deletion of this enhancer reduced MGMT expression (*Chen et al., 2018*). As BET transcription factors bind elevated H3K27ac levels found in enhancers (*Sengupta et al., 2015*; *Lovén et al., 2013*), this may suggest a possible mechanism for BETi-induced reduction of MGMT expression, which in turn results in increased sensitivity to the DNA alkylating agent TMZ.

Silencing the MGMT gene through promoter methylation has long been known to increase responsiveness to TMZ treatment and improve prognosis in patients with glioblastoma (*Karayan-Tapon et al., 2010*; *Hegi et al., 2005*; *Esteller et al., 2000*). Yet Despite that, clinical trials that combine TMZ and MGMT inhibitors have not improved therapeutic outcomes in such patients, possibly due to the 50% reduction in dose of TMZ, which is required to avoid hematologic toxicity (*Quinn et al., 2009a*; *Quinn et al., 2009b*; *Quinn et al., 2009c*). Thus, BETi offers an attractive line of research, though further studies are needed to determine whether the elevated sensitivity of glioblastoma to BETi, and its ability to reduce MGMT expression could be exploited to improve patient outcome.

## MIEL provides a reliable proxy of the transcriptomic profile

We analyzed serum and BMP4, two established biologicals known to induce glioblastoma differentiation in culture (*Lee et al., 2006*; *Piccirillo et al., 2006*; *Pollard et al., 2009*) and established signatures of the differentiated glioblastoma cells based on the pattern of epigenetic marks that could be applied across several genetic backgrounds. This is the first time that a signature for glioblastoma differentiation suitable for high-throughput drug screening has been obtained. Indeed, results of previous studies using bulk glioblastoma analysis (*Carén et al., 2015*) or single-cell sequencing (*Patel et al., 2014*) could not be readily applied for high-throughput screening. As a proof of principle, we analyzed the Prestwick chemical library (1200 compounds) to validate MIEL's ability to select and prioritize small molecules, which mimic the epigenetic and transcriptomic effects of serum and BMP4. Surprisingly, we observed that the degree of reduction in endogenous SOX2 protein levels following drug treatment did not correlate with the degree of differentiation assessed by global gene expression (*Figure 5—figure supplement 2d*); in contrast, MIEL-based metrics did correlate. This result, taken together with MIEL's ability to distinguish multiple cells types (iPSCs, NPCs, fibroblasts, hematopoietic lineages; *Figure 4b,c*; *Figure 4—figure supplement 2*) across several genetic backgrounds, demonstate that the MIEL approach can readily identify compounds inducing desired changes in cell fate and that it can serve as a cost-effective tool for prioritizing compounds during the primary screenings.

By tapping into the wealth of information contained within the cellular epigenetic landscape through modern high-content profiling and machine-learning techniques, the MIEL approach represents a valuable tool for high-throughput screening and drug discovery and is especially relevant when the desired cellular outcome cannot be readily defined using conventional approaches.

## Materials and methods

### Cell culture

Monolayer cultures of patient-derived GMB TPCs were propagated on Matrigel-coated plates in DMEM:F12 Neurobasal Medium (1:1; Gibco), 1% B27 supplement (Gibco), 10% BIT 9500 (StemCell Technologies), 1 mM glutamine, 20 ng/ml EGF (Chemicon), 20 ng/ml bFGF, 5 µg/ml insulin (Sigma), and 5 mM nicotinamide (Sigma). The medium was replaced every other day and the cells were enzymatically dissociated using Accutase prior to splitting. Fibroblasts, iPSCs, and iPSC-derived NPCs were cultured as previously described (*Marchetto et al., 2010*; *Kim et al., 2011*).

## Differentiation treatment

For TPC differentiation treatments cells were cultured in DMEM:F12 Neurobasal Medium (1:1), 1% B27 supplement, 10% BIT 9500, 1 mM glutamine supplemented with either Bmp4 (100 ng/ml; R and D Systems) or FBS (10%).

## Immunofluorescence

Cells were rinsed with PBS and fixed in 4% paraformaldehyde in PBS for 10 min at room temperature. After blocking with PBSAT (2% BSA and 0.5% Triton X-100 in PBS) for 1 hr at room temperature, the cells were incubated overnight at 4°C with primary antibodies diluted in PBSAT. Primary antibodies are listed in *Supplementary file 1* - Table S1, and the appropriate fluorochrome-conjugated secondary antibodies were used at 1:500 dilution. Nuclear co-staining was performed by incubating cells with either Hoechst-33342 or DAPI nuclear dyes.

## Microscopy and image analysis

For MIEL analysis, cells were imaged on either an Opera QEHS high-content screening system (PerkinElmer) using ×40 water immersion objectives or an IC200-KIC (Vala Sciences) using a × 20 objective. Images collected were analyzed using Acapella 2.6 (PerkinElmer). At least 40 fields/well for Opera and five fields/well for IC200 were acquired and at least two wells per population were used. Features of nuclear morphology, fluorescence intensity inter-channel co-localization, and texture features (Image moments, Haralick, Threshold Adjacency Statistics) were calculated using custom algorithms (Source Code File one and www.andrewslab.ca). A full list of the features used is available from the authors. Values for each cell were generated and exported to Microsoft Excel or MATLAB for further analysis. For Sall2, Olig2, Brn2, Sox2, Oct4, and GFAP immunostaining, images were captured on an IC200-KIC (Vala Sciences) using a × 20 objective. Between 3 and 8 fields per well were acquired and analyzed using Acapella 2.6 (PerkinElmer). For all nuclear markers, average intensities in nucleus or fold change compared to untreated cells are shown. Unless stated otherwise, at least three wells and a minimum of 300 cells for each condition were compared using the unpaired two-tailed t-test.

## Data processing

The image features-based profile for each cell population (e.g., cell types, treatments, technical repetition) was represented using a vector (center of distribution vectors) in which every element is the average value of all cells in that population for a particular feature. The vector's length is given by the number of features chosen (262 per histone modification). Raw feature values were normalized by z-scoring to the average and standard deviation of all populations being compared. All cells in each population were used to calculate center vectors, and each population contained at least 50 cells. Activity level for each drug was determined by calculating the distance from DMSO. For this, feature values of all DMSO replicates center vectors were used to calculate the DMSO center vector. Euclidean distance of each compound and each DMSO replicate to the DMSO center vector was calculated. Distances were z-scored to the average distance and standard deviation of DMSO replicates from the DMSO center vector. Transcriptomic-based profile for each cell population was represented using a vector in which every element is the z-scored FPKM value for a single gene in that population. The length of the vector is given by the number of genes used to construct the profile.

## Multidimensional scaling - MDS

The Euclidean distance between all vectors (either image features or transcriptomic based) was calculated to assemble a dissimilarity matrix (size N × N, where N is the number of populations being compared). For representation, the N × N matrix was reduced to a Nx2 matrix with MDS using the Excel add-on program Xlstat (Base, v19.06), and displayed as a 2D scatter plot.

## Discriminant Analysis

Quadratic discriminant analysis was conducted using the Excel add-on program xlstat (Base, v19.06). The model was generated in a stepwise (forward) approach using default parameters. All features derived from images of tested histone modification were used for analysis following normalization by z-score. Features displaying multicollinearity were reduced. Model training was done using

multiple DMSO replicates and at least two replicates from each cell-line or drug treatment. The model was tested on at least 8 DMSO replicates and at least one replicate from each cell line or treatment.

## SVM classification

SVM classification was conducted as previously described (*Collins et al., 2015*). Cell-level data in total populations (minimum 400 cells per population) were normalized to z-scores, and a subset of cells from each population being classified was randomly chosen as the training set (subset size at least 100 × the population number bei ng classified). The training set was used for a SVM classifier (MATLAB svmtrain function). The remaining cells (test set) were then classified using the SVM-derived classifier to assess the accuracy of classification (MATLAB svmclassify function). Here, the accuracy of all pairwise classifications was given as the average accuracy calculated for each population. To classify the similarity of multiple cell populations, we classified known populations (e.g., different treatments or cell fates) to generate known bins and then used the same classifiers on the unknown population to categorize each cell.

## Epigenetic drug screening

GBM2 cells were plated at 4000 cells/well and exposed to epigenetic compounds (*Supplementary file 1* - Table S2) at 10 µM for 1 day in 384-well optical bottom assay plates (Perki-nElmer). Negative control was DMSO (0.1%), 48 DMSO replicates per plate, three technical replicates (wells) were treated per compound. Cells were fixed and stained with histone modification-specific antibodies (H3K27ac and H3K27me3, H3K9me3, H3K4me1) and AlexaFluor-488- or Alexa-Fluor-555-conjugated secondary antibodies. DNA was stained with DAPI followed by imaging and feature extraction. To compare data from multiple plates, average feature values in each plate were normalized to DMSO. Here, feature values of all DMSO replicates center vectors in each plate, then were used to calculate the plate-wise DMSO vector. Raw feature values for all center vectors of all populations in each plate were normalized to the plate-wise DMSO vector; normalized feature values were z-scored as above. To identify active compounds, activity level for each compound was calculated as above, and active compounds were defined as compounds for which activity z-score was >3. Compounds reducing the number of imaged cells per well below 50 were considered toxic and excluded from analysis.

## Concentration curves

GBM2 cells were plated and stained as above. For each compound (*Supplementary file 1* - Table S3), cells were treated at 0.1, 0.3, 1.0, 3.0, 10.0 uM. Activity levels were calculated as above. Average cell count was calculated across the replicates for each compound to compare epigenetic changes and toxicity. Cell counts were z-scored against the average and standard deviation of all DMSO replicates. Distances (z-scored) and cell counts (z-scored) were averaged for each functional class at each concentration.

## RNAseq and transcriptomic analysis

Total RNA was isolated from GBM2 cells using the RNeasy Kit (Qiagen), 0.25 ug total RNA was used to isolate mRNAs and for library preparation. Library preparation and sequencing were conducted by the SBP genomics core (Sanford-Burnham NCI Cancer Center Support Grant P30 CA030199). PolyA RNA was isolated using the NEBNext Poly(A) mRNA Magnetic Isolation Module, and bar-coded libraries were made using the NEBNext Ultra II Directional RNA Library Prep Kit for Illumina (NEB, Ipswich MA). Libraries were pooled and single-end sequenced (1 × 75) on the Illumina Next-Seq 500 using the High-Output V2 kit (Illumina). Read data, processed in BaseSpace (https://base-space.illumina.com), were aligned to *Homo sapiens* genome (hg19) using STAR aligner (https://code.google.com/p/rna-star/) with default settings. Differential transcript expression was determined using the Cufflinks Cuffdiff package (https://github.com/cole-trapnell-lab/cufflinks). For heat maps showing fold change in expression, FPKM values in each HDACi-treated population were divided by the average FPKM values of DMSO-treated GBM2 and values shown as log2 of the ratio. Go enrichment analysis was conducted using PANTHER v11 (*Mi et al., 2017*) using all genes identified as differentially expressed following either serum or Bmp4 treatment. To highlight differences in

expression levels between serum- and Bmp4-treated GBM2 cells, FPKM values in each sample were z-scored. Zscore=(FPKM$_{Observation}$-FPKM$_{Average}$)/FPKM$_{SD}$ (FPKM$_{Observation}$- FPKM value obtain through sequencing; FPKM$_{Average}$ – average of all FPKM values in all samples for a certain gene; FPKM$_{SD}$ – standard deviation of FPKM values for a certain gene). Heat maps were generated using Microsoft Excel conditional formatting.

## Comparing epigenetic changes in different cell lines

To compare drug-induced epigenetic changes across multiple glioblastoma cell lines, 101A, 217M, GBM2 and PBT24 cells were plated at 4000 cells/well and treated with compounds for 24 hr. Compounds and concentrations are shown in *Supplementary file 1* - Table S4. Activity level was calculated as above. Pearson coefficient and significance of correlation for activity levels in each pair of cell lines were calculated using the Excel add-on program xlstat (Base, v19.06).

## Correlation of transcriptomic and image-based profiles

Euclidean distances were calculated using either transcriptomic data (FPKM) or texture features. Pearson's correlation coefficient (R) was transformed to a t-value using the formula (t = R $\times$ SQRT(N-2)/SQRT(1-R2) where N is the number of samples, R is Pearson correlation coefficient; the p-value was calculated using Excel t.dist.2t(t) function. For compound prioritization, Euclidean distance between the compound treated and serum- or Bmp4-treated GBM2 cells was calculated based on either FPKM)or image features. The average distance for both serum and Bmp4 treatments was normalized to the average distance of untreated cells to serum and Bmp4.

## Sensitization to radiation or TMZ

Cells were plated at 1500 cells/well in 384-well optical bottom assay plates (PerkinElmer). Two sets of the experiment were prepared; DMSO (0.1%) was used for negative controls at 48 DMSO replicates per plate; three replicates (wells) were treated per compound. Compound concentrations used are shown in *Supplementary file 1* - Table S5. Cells in both sets were pre-treated with epigenetic compounds for 2 days prior to cytotoxic treatment. Cytotoxic treatment, either 200 μM temozolomide (TMZ, Sigma) or 1Gy x-ray radiation (RS2000; RAD Source) was carried out for 4 days on single set ('treatment set'); for TMZ treatment, DMSO control was given to the second set. A single radiation dose was given at day 3; TMZ was given twice at days 3 and 5 of the experiment. Cells were fixed, stained with DAPI, and scored using an automated microscope (Celigo; Nexcelom Bioscience). For each compound, fold change in cell number was calculated for both the 'treatment set' (Drug+Cytotox) and the 'control set' (Drug), compared to DMSO-treated wells in the control set. The effect of radiation or TMZ alone was calculated as fold reduction of DMSO-treated wells in the treatment set compared to DMSO-treated wells in the control set (Cytotox). The coefficient of drug interaction (CDI) was calculated as (Drug+Cytotox)/ (Drug)X(Cytotox). For conformation experiments, the same regiment and CDI calculations were carried out on SK262, 101A, 217M, 454M, and PBT24 glioblastoma cell lines; PARPi and BETi were used at same concentration as the initial screen on GBM2 (Table S5).

## Prestwick chemical library screen using H3K27me3 and H3K27ac

GBM2 cells were plated at 2000 cells/well and exposed to Prestwick compounds (3 μM; *Supplementary file 1* - Table S6) for 3 days in 384-well optical bottom assay plates (PerkinElmer). Cells were then fixed and stained with rabbit polyclonal anti-H3K27ac and mouse monoclonal anti-H3K27me3 antibodies followed by AlexaFluor-488- or AlexaFluor-555-conjugated secondary antibodies. Positive controls contained BMP4 (100 ng/ml) and serum (10%); negative controls contained DMSO (0.1%). DNA was counterstained with Hoechst. Images were acquired using Perkin Elmer Opera QEHS. MIEL analysis was conducted as described above.

## Acknowledgements

We are thankful to Harley Kornblum (UCLA) for sharing multiple primary human glioblastoma lines, Alysson Muotri (UCSD) for providing fibroblast, iPSC and NPC lines, and Bradley Bernstein (MGH Harvard) for sharing MGG-TPCs and MGG-DGCs lines, Laure Escoubet (Celgene) for discussions and

support, Alex Kiselyov (Genea Biocells) for help with initial compound libraries and discussions. We owe a debt of gratitude to Susanne Heynen-Genel, Debbie Chen, and other members of the High-Content Facility at CPCCG for their invaluable help with cell imaging and to Brian James and Kang Liu at the SBP Genomics core for their help with library preparation and RNA sequencing (NCI Cancer Center Support Grant P30 CA030199). We thank Linda Penn for suggestions and help with the manuscript. This work was supported by sponsored research agreement with Celgene to AVT, an R01 NS066278 to AVT, and by a CIHR Foundation grant and a Tier 1 Canada Research Chair award to DWA.

## Additional information

### Funding

| Funder | Grant reference number | Author |
|---|---|---|
| California Institute for Regenerative Medicine | TG2-01162 | Chen Farhy |
| Celgene | SCRA | Alexey V Terskikh |
| National Institutes of Health | R01 NS066278 | Alexey V Terskikh |
| Canada Research Chairs | Tier 1 Canada Research Chair | David W Andrews |
| Canadian Institutes of Health Research | | David W Andrews |

The funders had no role in study design, data collection and interpretation, or the decision to submit the work for publication.

### Author contributions

Chen Farhy, Conceptualization, Resources, Data curation, Formal analysis, Validation, Visualization, Methodology, Writing—original draft, Writing—review and editing; Santosh Hariharan, Jarkko Ylanko, Software; Luis Orozco, Fu-Yue Zeng, Ian Pass, Fernando Ugarte, E Camilla Forsberg, Chun-Teng Huang, Resources; David W Andrews, Conceptualization, Data curation, Software, Supervision, Methodology, Writing—original draft; Alexey V Terskikh, Conceptualization, Resources, Data curation, Supervision, Funding acquisition, Visualization, Methodology, Writing—original draft, Writing—review and editing

### Author ORCIDs

Chen Farhy ⒾⒹ https://orcid.org/0000-0001-6160-3479
David W Andrews ⒾⒹ http://orcid.org/0000-0002-9266-7157
Alexey V Terskikh ⒾⒹ https://orcid.org/0000-0003-4641-3997

### Decision letter and Author response

Decision letter https://doi.org/10.7554/eLife.49683.sa1
Author response https://doi.org/10.7554/eLife.49683.sa2

## Additional files

### Supplementary files

• Source code 1. Acapella script used for image processing.

• Supplementary file 1. Tables of resources.

• Transparent reporting form

### Data availability

Sequencing data have been deposited in GEO under accession code GSE134045.

The following dataset was generated:

| Author(s) | Year | Dataset title | Dataset URL | Database and Identifier |
|---|---|---|---|---|
| Farhy C, Terskikh A | 2019 | Improving drug discovery using image-based multiparametric analysis of the epigenetic landscape | https://www.ncbi.nlm.nih.gov/geo/query/acc.cgi?&acc=GSE134045 | NCBI Gene Expression Omnibus, GSE134045 |

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
