## [Decision Letter]

[Editors’ note: this article was originally rejected after discussions between the reviewers, but the authors were invited to resubmit after an appeal against the decision.]

Thank you for submitting your work entitled "Improving drug discovery using image-based multiparametric analysis of the epigenetic landscape" for consideration by *eLife*. Your article has been reviewed by two peer reviewers, and the evaluation has been overseen by a Reviewing Editor and a Senior Editor. The reviewers have opted to remain anonymous.

Our decision has been reached after consultation between the reviewers. Based on these discussions and the individual reviews below, we regret to inform you that your work will not be considered further for publication in *eLife*.

Although the reviewers found aspects of the work to be of merit, there were substantive concerns relating to the overall impact of the new technology and the broad relevance of this technique to drug discovery. Despite interest in the work, these concerns were substantive enough to preclude publication in *eLife*.

Reviewer #1:

In this manuscript, Farhy and coworkers developed an image-based analysis to examine epigenetic landscapes. This method was developed from a prior multivariate image analysis (Collins et al., 2015) in the context of epigenetic modulation. Here the core idea is to rely on a set of structural features of epigenetic marks (H3K27me3, H3K9me3, H3K27ac and H3K4me) as readouts of epigenetic modulation. These structural features were extracted via machine-learning algorithms described previously. To demonstrate the utility of this method, the authors first showed that the current approach is more sensitive than prior "intensity"-based approaches to detect the effects of small-molecule epigenetic modulators. The authors then showed that the multi-variate readouts correlate with treatment doses and the compounds sharing common targets in the context of multiple cell lines. With this approach, the authors further identified the epigenetic modulators that synergistically work with TMZ and radiation and distinguished the cell fates/differentiation of several types of cells. Novelty of this work mainly lies in the machine-learning-based algorithm to extract a set of structural features of chromatin as readouts. However, the reviewer feels that this work is largely an incremental data analysis method with multiple limitations as detailed below.

1) To examine epigenetic modulation, many robust approaches such as ATAC-seq/ChIP-seq have been developed as noted by the authors. Their methods provide rich information with terrific resolution. In contrast, various Western-blot assays or image-intensity-based assays have been developed to examine the global levels of certain epigenetic marks (H3K27me3, H3K9me3, H3K27ac and H3K4me). Merits of the MIEL largely lies between the two conventional approaches. However, the challenge of data collections (three or four sets of data for a treatment condition) and the complexity of data processing (feature extraction and combination) significantly limited the broad use of this approach.

2) In contrast with the conventional image-intensity-based assays, MIEL only revealed around 10 additional compounds among >200 candidates that alter the epigenetic landscape (data of Figure 1B). However, it is not clear about the mechanism that the "10" additional compounds cannot be identified with the conventional image-intensity-based assays but can be revealed with MIEL. It is not clear how the authors ruled out the possibility that the changes of epigenetic landscapes associated with the indirect outcomes of off-target effects of the 10 compounds. If it is the case, the positive readouts become irreverent to the primary targets of these compounds.

3) The underlying molecular mechanism of MIEL is not clear. The altered structural features are very descriptive and may not link to perturbation in a casual manner. As a result, it is likely that the effects of many epigenetic modulators may not be readily detected with MIEL; the MIEL-detectable changes may not be directly relevant to epigenetic biology. As a result, MIEL needs to be fully validated before its implementation in a specific context. Meanwhile, ATAC-seq/ChIP-seq, Western-blot assays and image-intensity based assays are often developed on the basis of molecular mechanisms of epigenetic modulation. The latter are thus more robust given the direct causal relationship between the targets of interest and their function-related readouts.

In short, in the context of many alternative conventional approaches, the complexity and limitations of MIEL outweigh its merits. The reviewer doesn't feel that the quality of this manuscript reaches the standard of *eLife* in terms of novelty and broad impact.

Reviewer #2:

Chen Farhy and colleagues report on an elegant high-throughput way to use image analysis that can identify epigenetically active drugs, classify them by molecular function, and assess candidate drugs for their ability to increase sensitivity to chemotherapeutic agents. This is a really understudied area that has traditionally required expensive plate reader based tools to quantify. The manuscript is very detailed and thorough and has addressed the utility of the MIEL tool in a wide range of drug screening settings.

The two major concerns are the lack of certain control conditions necessary to support the authors claims.

1) The authors call their phenotypic screening platform "Microscopic Imaging of Epigenetic Landscape (MIEL)", and the imaging is solely based on texture features of four immunolabeled epigenetic marks. However, the texture of those histone modifications can be correlated to the general structure and texture of the DNA. The authors exclude compounds that lead to a cell count > 50 nuclei/ well, however, if a compound induces significant apoptosis with ~50% cell death, the texture features could pick up chromatin condensation and thereby potentially generate false-positive hits.

It would be helpful to see if there is a correlation between nuclei count and MIEL z-scores.

Furthermore, the authors claim that they can use MIEL to analyse dose-dependent effects from drug treatment. Yet, in light of the concern above, they cannot be sure whether they detect the pharmacological effect or a toxic effect that is not related to epigenetic changes. Concerning is the fact that the initial screen was conducted for just 24h and still was sufficient to separate the drug classes in clusters. However, later in the manuscript the authors increased the treatment times to 2 or 3 days, which reflects much more the time frame required to induce detectable epigenetic changes.

It would be great if the authors could put a few cytotoxic drugs through their pipeline that act not via epigenetic mechanisms but rather are inducers of apoptosis, necrosis, DNA damage, or cell-cycle arrest, and see whether they cluster with any of the epigenetic modifier classes or form separate clusters.

2) The second major concern is that, for many of the analyses it seems almost irrelevant whether all four histone modifications are taken into account, or just either of the two pairs, or just one of the four marks. This could be an indicator, that the texture features that are being extracted are not specific to the histone modifications, but rather general changes to DNA structure.

To clarify this, it would be recommended to stain just for DNA structure (DAPI) and overall Histone structure (H3), treat with representative compounds of the drug screen, and extract the same/ comparable texture features that were used for MIEL. Using DAPI and H3 texture, is it then also possible to discriminate between the compounds?

The second experiment to confirm that the changes in texture are due to a change in Histone modification landscape and not due to non-specific alteration of DNA structure, is to add a second detection method for the specific Histone modifications. ATAC sequencing after treatment with and without representative compounds of the drug screen should provide biological evidence for the phenotypic results of the image analysis.

---

## [Author Response]

[Editors’ note: the author responses to the first round of peer review follow.]

Reviewer #1:In this manuscript, Farhy and coworkers developed an image-based analysis to examine epigenetic landscapes. This method was developed from a prior multivariate image analysis (Collins et al., 2015) in the context of epigenetic modulation. Here the core idea is to rely on a set of structural features of epigenetic marks (H3K27me3, H3K9me3, H3K27ac and H3K4me) as readouts of epigenetic modulation. These structural features were extracted via machine-learning algorithms described previously. To demonstrate the utility of this method, the authors first showed that the current approach is more sensitive than prior "intensity"-based approaches to detect the effects of small-molecule epigenetic modulators. The authors then showed that the multi-variate readouts correlate with treatment doses and the compounds sharing common targets in the context of multiple cell lines. With this approach, the authors further identified the epigenetic modulators that synergistically work with TMZ and radiation and distinguished the cell fates/differentiation of several types of cells. Novelty of this work mainly lies in the machine-learning-based algorithm to extract a set of structural features of chromatin as readouts. However, the reviewer feels that this work is largely an incremental data analysis method with multiple limitations as detailed below.1) To examine epigenetic modulation, many robust approaches such as ATAC-seq/ChIP-seq have been developed as noted by the authors. Their methods provide rich information with terrific resolution. In contrast, various Western-blot assays or image-intensity-based assays have been developed to examine the global levels of certain epigenetic marks (H3K27me3, H3K9me3, H3K27ac and H3K4me). Merits of the MIEL largely lies between the two conventional approaches.

We appreciate reviewer’s concerns regarding the limitations of the MIEL platform. Indeed, ATAC-seq/ChIP-seq provide terrific resolution when a limited number of experimental conditions are studied, and we do not intend for MIEL to be compared with ATAC-seq/ChIP-seq in this setting. However, ATAC-seq/ChIP-seq are not amenable (either technically or economically) for high-throughput drug screening campaigns, which involve several hundred cells per analysis and several hundred thousands of compounds per screening. In contrast, the MIEL platform is intended to be employed for high-throughput phenotypic screening to improve drug discovery. ATAC-seq/ChIP-seq and MIEL platforms would thus complement, rather than compete with, each other by addressing different needs and applications.

The comparison to image-intensity-based assays is indeed warranted, and our data extensively document the remarkable superiority of MIEL over the intensity-based assays (Figure 1B; Figure 1—figure supplement 2A, B; Figure 1—figure supplement 3A).

However, the challenge of data collections (three or four sets of data for a treatment condition) and the complexity of data processing (feature extraction and combination) significantly limited the broad use of this approach.

The challenges of data collection are similar for the intensity-based assays and MIEL because even one channel acquisition is sufficient for both; however, single-channel MIEL is still far superior to the intensity-based assays (Figure 1—figure supplement 2A, B). In fact, we would argue that data collection is easier with MIEL, because a common standard set of epigenetic marks is used for all types of samples and analyses, and the specific antibodies could be prepared and tested in advance for all subsequent assays. As for the complexity of data processing (feature extraction and combination), we employ a previously described pipeline (Collins et al., 2015; and Oppermann et al., 2016), which is entirely automated, has a step-by-step manual, requires no computation or mathematics training, and is routinely executed by technical personnel in our laboratory. Furthermore, similar software packages are available that do not rely on pre-written scripts but rather utilize an easy-to-use building-block interface (e.g., CellProfiler).

While we appreciate that in many fields it could be difficult to interpret the mechanism behind the models constructed through machine-learning algorithms, the utility of such approaches is widely recognized and they represent the leading edge in the current drug discovery process.

2) In contrast with the conventional image-intensity-based assays, MIEL only revealed around 10 additional compounds among >200 candidates that alter the epigenetic landscape (data of Figure 1B).

We apologize for insufficient clarity in our explanation, which has led to this misunderstanding. First, it is important to note that the power of the MIEL platform lies in its ability to identify the multiparametric signature of epigenetically active compounds and thus accurately classify such compounds according to their mechanism of action, as demonstrated in our study (Figure 1C, D). In contrast, we showed that intensity-based assays lack such classification power (Figure 1—figure supplement 3A). It is predominantly this advantage of MIEL over intensity-based analysis that we are aiming to highlight.

We apologize that we did not clearly state the different number of compounds identified by MIEL vs. intensity-based analyses in the original manuscript. MIEL identified 122 active compounds that induced significant epigenetic changes, whereas the intensity-based approach identified 94 active compounds using the same stringent criteria (outside ± 3 z-scores). The additional 28 compounds identified by MIEL cover virtually all functional categories of epigenetic activity (Figure 1B and Figure 1—figure supplement 2).

However, it is not clear about the mechanism that the "10" additional compounds cannot be identified with the conventional image-intensity-based assays but can be revealed with MIEL. It is not clear how the authors ruled out the possibility that the changes of epigenetic landscapes associated with the indirect outcomes of off-target effects of the 10 compounds. If it is the case, the positive readouts become irreverent to the primary targets of these compounds.

The reviewer raises the possibility that these 28 additional compounds identified as active by MIEL, but not by intensity-based analysis, elicit off-target effects. Of the 122 compounds that were identified as active by MIEL, 85 were used for functional classification analysis (Figure 1C, D; the remaining 37 compounds belonged to functional classes with an insufficient number of active compounds to support classification algorithms). Of these 85 compounds, 10 were detected as active by MIEL but not by intensity-based analysis, and 8 of these 10 compounds (80%) were correctly classified to the expected functional category based on their known mechanism of action. Therefore, if these compounds do induce off-target effects, they are not dominant and do not preclude correct classification.

We believe that MIEL’s ability to register off-target effects should be considered as an advantage, enabling a measure of drug specificity in addition to drug activity. Both academic labs and pharmaceutical companies (e.g., Recursion Pharmaceuticals) have identified and successfully exploited off-target effects of compounds. For instance, we identified distinct concentration-dependent patterns for some epigenetic compounds, suggesting possible divergent cell fate outcomes at low vs. high concentrations (Figure 2A). We suggest that this ability further supports the relevance and superiority of the MIEL pattern-based analysis compared with intensity-based readouts.

3) The underlying molecular mechanism of MIEL is not clear. The altered structural features are very descriptive and may not link to perturbation in a casual manner. As a result, it is likely that the effects of many epigenetic modulators may not be readily detected with MIEL; the MIEL-detectable changes may not be directly relevant to epigenetic biology. As a result, MIEL needs to be fully validated before its implementation in a specific context. Meanwhile, ATAC-seq/ChIP-seq, Western-blot assays and image-intensity based assays are often developed on the basis of molecular mechanisms of epigenetic modulation. The latter are thus more robust given the direct causal relationship between the targets of interest and their function-related readouts.

We apologize for the lack of clarity in our explanation of the relationship between the MIEL-detectable changes and epigenetic biology. The mechanism of MIEL analysis is deeply rooted in the topology of the cellular epigenome and thus causally linked to perturbations of the epigenome. For example, changing the cell fate (differentiation) involves specific changes in the open/closed chromatin loci, which until now have been studied in bulk, averaging thousands or more cellular states. In contrast, MIEL is capable of assessing the epigenetic landscape in single cells. The sensitivity of MIEL approach, namely, the minimum change of epigenetic landscape that can be detected by MIEL in a single cell is, indeed, a key question and can only be answered experimentally.

Our present study is focused on addressing this very question in great detail using three specific contexts. Namely, we have validated MIEL’s utility and power (1) to detect and cluster diverse compounds using a large collection of established epigenetic drugs, (2) to distinguish different stages of cellular differentiation and identify signatures of such stages using a primary human glioblastoma model, and (3) to inform the mechanism of drug function using a model of TMZ synergy with PARP or BRD inhibitors. We verified the accuracy of MIEL-based readouts for these three specific contexts using previously published data or whole genome sequencing. Such experiments provide direct validation of MIEL in these contexts.

As a testimony to the breakthrough that we have achieved, we note that none of the applications demonstrated in the present manuscript was previously possible using a single universal set of markers applicable to any mammalian cell. Thus, the generalizability of our approach surpasses all previous ones. Moreover, unlike “…traditional cell painting approaches rely on unspecified distribution of arbitrary markers” (Pennisi E. Cell painting highlights responses to drugs and toxins. Science 2015:352;877-878), our approach directly interrogates the nuclear distribution (pattern) of epigenetic marks. Such epigenetic marks and their distribution have a well-defined biological meaning. For instance, H3K9me3 mark largely demarcates close/inactive chromatin, while H3K9ac marks labels accessible/expressed chromatin.

Indeed, the topography of epigenetic marks in the nucleus is directly related to the topology of active/inactive chromatin. Epigenetic marks are universal determinants of cellular fate and are present in all eukaryotic cells. Our discovery that these marks provide unique and importantly, interpretable information about the epigenetic landscape of single cells is not an incremental advance. Instead, it led to the novel insight that similar distributions of nuclear epigenetic marks obtained from image-based analysis equate with similar cellular identity, at least in the several examples described in our manuscript. This contrasts with many conventional phenotypic screening configurations, where cells with similar “distribution of arbitrary markers” have no mechanistic basis to equate with similar cellular identity.

In short, in the context of many alternative conventional approaches, the complexity and limitations of MIEL outweigh its merits. The reviewer doesn't feel that the quality of this manuscript reaches the standard of eLife in terms of novelty and broad impact.Reviewer #2:[…] The two major concerns are the lack of certain control conditions necessary to support the authors claims.1) The authors call their phenotypic screening platform "Microscopic Imaging of Epigenetic Landscape (MIEL)", and the imaging is solely based on texture features of four immunolabeled epigenetic marks. However, the texture of those histone modifications can be correlated to the general structure and texture of the DNA. The authors exclude compounds that lead to a cell count > 50 nuclei/ well, however, if a compound induces significant apoptosis with ~50% cell death, the texture features could pick up chromatin condensation and thereby potentially generate false-positive hits.

As suggested by the reviewer, karyopyknosis is a major concern that would inevitably have a detrimental effect on the staining texture. Fortunately, pyknotic nuclei are very distinct from normal nuclei, even at early stages of pyknosis, and can be easily excluded from the analysis during the nuclei segmentation stage (Author response image 1; red arrow showing unsegmented pyknotic nucleus).

**Author response image 1. respfig1:** Image of DAPI-stained mouse hepatocytes showing segmentation of normal healthy nuclei (green arrows), but not of pyknotic or pre-pyknotic nuclei (red arrow).

It would be helpful to see if there is a correlation between nuclei count and MIEL z-scores.

We agree with the reviewer that it is important to identify a possible correlation between drug toxicity (cell count) and magnitude of the effect (z-scored Euclidean distance) because it may suggest that cell stress is a major contributor to the staining textures analyzed by MIEL. We explored this question in the manuscript (Figure 2B) where we report that: “… some compound classes, such as Aurora and JAK inhibitors, induce epigenetic alterations only in concentrations at which the cell count is significantly reduced, whether through toxicity or a direct effect on proliferation (original Figure 2B, dark blue and pink, respectively), while other compounds, such as HDAC inhibitors, characteristically induce epigenetic alterations at concentrations that are not accompanied by reduced cell counts (green and yellow). Interestingly, inhibitors of both SIRT and EZH1/2 (light blue and red, respectively) induced significant epigenetic changes without significantly affecting cell count.”

Furthermore, the authors claim that they can use MIEL to analyse dose-dependent effects from drug treatment. Yet, in light of the concern above, they cannot be sure whether they detect the pharmacological effect or a toxic effect that is not related to epigenetic changes.

The reviewer’s concerns are fully justified. However, as shown above, some drug classes (e.g., EZH1/2 inhibitors) can induce epigenetic changes without toxicity. In addition, we offer Author response image 2,which shows a distance map of GBM2 cells treated with DMSO or HDAC inhibitors at multiple concentrations (the size of each node is proportional to the cell count of each population). Treatments that reduced cell counts are partial ‘outliers’ compared with the center of the HDAC ‘cloud’ (green arrows), and a significant reduction in cell count (<50) tends to shift the compound completely away from the cloud (red arrow) Author response image 2. This observation further suggests that, while toxicity and cell stress play a role in shaping the epigenetic landscape, the staining texture remains sufficiently robust to correctly classify the treatment by drug function, at least up to a point.

**Author response image 2. respfig2:** Distance map depicting the relative Euclidean distance between the multiparametric centroids of GBM2 cells treated with DMSO (88 replicates) or pan HDAC inhibitors (7 compounds; average or 2 replicates shown). Compounds were present at 0.3, 1, 3, 10, and 30 μM. Distances calculated using texture features derived from images of cells stained with H3K9me3 and H3K27ac.

Concerning is the fact that the initial screen was conducted for just 24h and still was sufficient to separate the drug classes in clusters. However, later in the manuscript the authors increased the treatment times to 2 or 3 days, which reflects much more the time frame required to induce detectable epigenetic changes.

The time required to elicit epigenetic changes and the magnitude of such changes greatly depends on the compound, its concentration, and the targeted

histone modification (Zee BM et al. J Biol Chem. 2010). In the first set of experiments, we targeted epigenetic enzymes and incubated the cells with compounds for 24 hours in order to detect direct changes in histone modifications. Many of these compounds have been previously shown to act

quickly and to elicit a significant change within hours of treatment (e.g., Luense et al., 2015). In the later set of experiments on the differentiation of GBM TPCs, we used a longer treatment (3 days) to allow the cells to acquire as

many epigenetic changes as possible while maintaining a time frame suitable for drug screening.

It would be great if the authors could put a few cytotoxic drugs through their pipeline that act not via epigenetic mechanisms but rather are inducers of apoptosis, necrosis, DNA damage, or cell-cycle arrest, and see whether they cluster with any of the epigenetic modifier classes or form separate clusters.

We agree with this reviewer that it would be informative to understand whether MIEL can assign a compound to a biological category in cells that are exposed to toxic compounds but have not yet displayed the signs of pyknosis. Author response image 3 shows a distance map that includes GBM cells treated with all epigenetic compounds, including those excluded from the analyses shown in the manuscript (<50 nuclei). The distance map shows wide scattering of the various conditions leading to cell death (red). Our interpretation of this finding is that treatments that induce cell death do not converge to form a distinctive staining texture. This may be because the texture of dying cells is greatly influenced by both cell death-related processes and the mechanism of action of the toxic drug. In the future studies, it will be informative to compare the effect of drugs that are known to specifically induce apoptosis, necrosis, paraptosis, and other forms of cell death to further elucidate this question.

**Author response image 3. respfig3:** Distance map depicting the relative Euclidean distance between the multiparametric centroids of GBM2 cells treated with DMSO or 85 active compounds (3 technical replicates per compound; 91 DMSO replicates). Distances calculated using texture features derived from images of cells stained for H3K9me3, H3K27me3, H3K4me1, and H3K27ac.

2) The second major concern is that, for many of the analyses it seems almost irrelevant whether all four histone modifications are taken into account, or just either of the two pairs, or just one of the four marks. This could be an indicator, that the texture features that are being extracted are not specific to the histone modifications, but rather general changes to DNA structure.To clarify this, it would be recommended to stain just for DNA structure (DAPI) and overall Histone structure (H3), treat with representative compounds of the drug screen, and extract the same/ comparable texture features that were used for MIEL. Using DAPI and H3 texture, is it then also possible to discriminate between the compounds?

We thank the reviewer for his/her valuable contribution. We have previously examined the texture of DAPI-stained cells and found that DNA labeling can partially recapitulate the staining pattern of H3K9me3, which labels constitutive heterochromatin (Author response image 4; image processed to highlight the similarities). We have also conducted the analysis suggested by the reviewer and found that DAPI staining provided an overall classification accuracy of 65.6% (Author response image 4) compared with 86.4% provided by H3K9me3 (Author response image 4). However, despite the reduced overall accuracy, it is evident that DAPI staining may be an informative and cheap alternative to histone staining in at least some applications when the analyzed epigenetic landscape are very distinct. This analysis has been added to Figure 1—figure supplement 3B together with accompanying text in the Results section.

**Author response image 4. respfig4:** DAPI images can be used for MIEL analysis. (**A**) Image of DAPI-stained GBM2 cells demonstrating the similarities in DAPI and H3K9me3 staining (contrast and brightness adjusted to highlight similarity of stain). (**B, C**) Quadratic discriminant analysis using texture features derived from images of GBM2 cells treated with DMSO or 85 active compounds (2 technical replicates per compound; 38 DMSO replicates) stained for (**B**) DAPI or (**C**) H3K9me3 (reproduced from manuscript Figure 1—figure supplement 3B). Confusion matrices show classification results from discriminant analysis. Numbers represent the percentage compounds classified correctly (diagonal) and incorrectly (off diagonal).

The second experiment to confirm that the changes in texture are due to a change in Histone modification landscape and not due to non-specific alteration of DNA structure, is to add a second detection method for the specific Histone modifications. ATAC sequencing after treatment with and without representative compounds of the drug screen should provide biological evidence for the phenotypic results of the image analysis.

As suggested by the reviewer, it is very likely that information obtained by MIEL on the epigenetic landscape is at least partly influenced by both the organization of chromatin and the pattern of histone modifications. Although we found that one histone modification was as good as another for both the detection and classification of many treatments, this was not always the case. Perhaps the most obvious example is H3K27me3, which is relatively poor at function-based classification for all epigenetic drugs except EZH1/2 inhibitors, for which it yielded the highest classification accuracy of all modifications (Figure 1—figure supplement 3B of the manuscript). We believe that this is due to the direct role of EZH1/2 in the methylation of Lys27 of Histone 3.

We agree with the reviewer that ATAC-seq may add another layer of confidence to our results. We believe, however, that changes in overall staining intensity for various histone modifications (Figure 1—figure supplement 2B) together with changes in the transcriptomic program (validated using RNA-seq) are sufficiently strong to establish that epigenetic changes do indeed occur following drug, serum, or Bmp4 treatment.